# ADVERSARIAL CAUSAL BAYESIAN OPTIMIZATION

**Scott Sussex**
ETH Zürich
scott.sussex@inf.ethz.ch

**Pier Giuseppe Sessa**
ETH Zürich

**Anastasiia Makarova**
ETH Zürich

**Andreas Krause**
ETH Zürich

## ABSTRACT

In Causal Bayesian Optimization (CBO), an agent intervenes on a structural causal model with known graph but unknown mechanisms to maximize a downstream reward variable. In this paper, we consider the generalization where other agents or external events also intervene on the system, which is key for enabling adaptiveness to non-stationarities such as weather changes, market forces, or adversaries. We formalize this generalization of CBO as *Adversarial Causal Bayesian Optimization (ACBO)* and introduce the first algorithm for ACBO with bounded regret: *Causal Bayesian Optimization with Multiplicative Weights* (CBO-MW). Our approach combines a classical online learning strategy with causal modeling of the rewards. To achieve this, it computes optimistic counterfactual reward estimates by propagating uncertainty through the causal graph. We derive regret bounds for CBO-MW that naturally depend on graph-related quantities. We further propose a scalable implementation for the case of combinatorial interventions and submodular rewards. Empirically, CBO-MW outperforms non-causal and non-adversarial Bayesian optimization methods on synthetic environments and environments based on real-word data. Our experiments include a realistic demonstration of how CBO-MW can be used to learn users' demand patterns in a shared mobility system and reposition vehicles in strategic areas.

## 1 INTRODUCTION

How can a scientist efficiently optimize an unknown function that is expensive to evaluate? This problem arises in automated machine learning, drug discovery and agriculture. *Bayesian optimization* (BO) encompasses an array of algorithms for sequentially optimizing unknown functions (Močkus, 1975). Classical BO algorithms treat the unknown function mostly as a black box and make minimal structural assumptions. By incorporating more domain knowledge about the unknown function into the algorithm, one can hope to optimize the function using fewer evaluations.

A recent line of work on causal Bayesian optimization (CBO) (Aglietti et al., 2020) attempts to integrate use of a structural causal model (SCM) into BO methods. It is assumed that actions are interventions on some set of observed variables, which are causally related to each other and a reward variable through a known causal graph (Fig. 1b), but unknown mechanisms. Many important BO problems might take such a shape. For example, managing supply in a Shared Mobility System (SMS) involves intervening on the distribution of bikes and scooters across the city. Importantly, Sussex et al. (2022) show that a BO approach leveraging the additional structure of CBO can achieve exponentially lower regret in terms of the action space size.

Most CBO methods to date assume that the system is completely stationary across interactions and that only one agent interacts with the system. However, often it would be desirable to incorporate the influence of external events. For example, in a SMS the day's demand is highly non-stationary, and can only be fully observed at the day's end. We would like an algorithm that *adapts* to the variability in these external events.

In this work, we incorporate external events into CBO by introducing a novel *adversarial* CBO (ACBO) setup, illustrated in Fig. 1c. Crucially, in ACBO the downstream reward is explicitly influenced by potentially adversarial interventions on certain nodes in the causal graph (identified using dashed nodes in Fig. 1c) that can only be observed a-posteriori. For this general setting, we propose a novel algorithm – CBO with multiplicative weights (CBO-MW) – and prove a regret

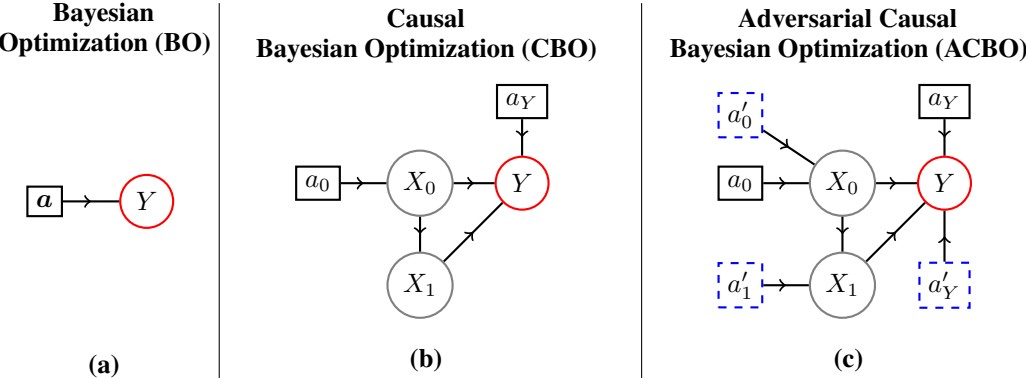

Figure 1: Visualizations of Bayesian Optimization (BO), Causal BO (CBO), and the Adversarial CBO (ACBO) introduced in this work. Agents must select actions $\boldsymbol{a}$ that maximize rewards $Y$. **(a)** Standard BO assumes the simplest DAG from actions to reward, regardless of the problem structure. **(b)** CBO incorporates side observations, e.g., $X_0, X_1$ and causal domain knowledge. Each observation could be modeled with a separate model. **(c)** In ACBO (this work), we additionally model the impact of *external events* (weather, perturbations, other players' actions, etc.) *that cannot be controlled*, but can only be observed a-posteriori. These are depicted as dashed blue nodes and could directly affect the reward node, like $a'_Y$, or indirectly affect it by perturbing upstream variables, like $a'_0, a'_1$.

guarantee using a stronger (but natural) notion of regret than the one used in existing CBO works. For settings where the number of intervention targets is large, we propose a distributed version of CBO-MW which is computationally efficient and can achieve approximate regret guarantees under some additional submodularity assumptions on the reward. Finally, we find empirical evidence that CBO-MW outperforms relevant prior work in adversarial versions of previously studied CBO benchmarks and in learning to re-balance units in an SMS simulation based upon real data.

## 2 BACKGROUND AND PROBLEM STATEMENT

We consider the problem of an agent interacting with an SCM for $T$ rounds in order to maximize the value of a reward variable. We start by introducing SCMs, the soft intervention model used in this work, and then define the adversarial sequential decision-making problem we study. In the following, we denote with $[m]$ the set of integers $\{0, \dots, m\}$.

**Structural Causal Models** Our SCM is described by a tuple $\langle \mathcal{G}, Y, \boldsymbol{X}, \boldsymbol{F}, \boldsymbol{\Omega} \rangle$ of the following elements: $\mathcal{G}$ is a *known* DAG; $Y$ is the reward variable; $\boldsymbol{X} = \{X_i\}_{i=0}^{m-1}$ is a set of observed scalar random variables; the set $\boldsymbol{F} = \{f_i\}_{i=0}^{m}$ defines the *unknown* functional relations between these variables; and $\boldsymbol{\Omega} = \{\boldsymbol{\Omega}_i\}_{i=0}^{m}$ is a set of independent noise variables with zero-mean and known distribution. We use the notation $Y$ and $X_m$ interchangeably and assume the elements of $\boldsymbol{X}$ are topologically ordered, i.e., $X_0$ is a root and $X_m$ is a leaf. We denote with $pa_i \subset \{0, \dots, m\}$ the indices of the parents of the $i$th node, and use the notation $\boldsymbol{Z}_i = \{X_j\}_{j \in pa_i}$ for the parents this node. We sometimes use $X_i$ to refer to both the $i$th node and the $i$th random variable.

Each $X_i$ is generated according to the function $f_i : \mathcal{Z}_i \to \mathcal{X}_i$, taking the parent nodes $\boldsymbol{Z}_i$ of $X_i$ as input: $x_i = f_i(\boldsymbol{z}_i) + \omega_i$, where lowercase denotes a realization of the corresponding random variable. The reward is a scalar $x_m \in [0, 1]$ while observation $X_i$ is defined over a compact set $x_i \in \mathcal{X}_i \subset \mathbb{R}$, and its parents are defined over $\mathcal{Z}_i = \prod_{j \in pa_i} \mathcal{X}_j$ for $i \in [m-1]$.[1]

**Interventions** In our setup, an agent and an adversary both perform *interventions* on the SCM [2]. We consider a soft intervention model (Eberhardt and Scheines, 2007) where interventions are parameterized by controllable *action variables*. A simple example of a soft intervention is a shift intervention, where actions affect their outputs additively (Zhang et al., 2021).

---

[1]Here we consider scalar observations for ease of presentation, but we note that the methodology and analysis can be easily extended to vector observations as in Sussex et al. (2022)

[2]Our framework allows for there to be potentially multiple adversaries, but since we consider everything from a single player's perspective, it is sufficient to combine all the other agents into a single adversary.

First, consider the agent and its action variables $\boldsymbol{a} = \{a_i\}_{i=0}^m$. Each action $a_i$ is a real number chosen from some finite set. That is, the space $\mathcal{A}_i$ of action $a_i$ is $\mathcal{A}_i \subset \mathbb{R}_{[0,1]}$ where $|\mathcal{A}_i| = K_i$ for some $K_i \in \mathbb{N}$. Let $\mathcal{A}$ be the space of all actions $\boldsymbol{a} = \{a_i\}_{i=0}^m$. We represent the actions as additional nodes in $\mathcal{G}$ (see Fig. 1): $a_i$ is a parent of only $X_i$, and hence an additional input to $f_i$. Since $f_i$ is unknown, the agent does not know apriori the functional effect of $a_i$ on $X_i$. Not intervening on a node $X_i$ can be considered equivalent to selecting $a_i = 0$. For nodes that cannot be intervened on by our agent, we set $K_i = 1$ and do not include the action in diagrams, meaning that without loss of generality we consider the number of action variables to be equal to the number of nodes $m$. [3]

For the adversary we consider the same intervention model but denote their actions by $\boldsymbol{a}'$ with each $a_i'$ defined over $\mathcal{A}_i' \subset \mathbb{R}_{[0,1]}$ where $|\mathcal{A}_i'| = K_i'$ and $K_i'$ is not necessarily equal to $K_i$.

According to the causal graph, actions $\boldsymbol{a}, \boldsymbol{a}'$ induce a realization of the graph nodes:

$$x_i = f_i(\boldsymbol{z}_i, a_i, a_i') + \omega_i, \quad \forall i \in [m]. \tag{1}$$

If an index $i$ corresponds to a root node, the parent vector $\boldsymbol{z}_i$ denotes an empty vector, and the output of $f_i$ only depends on the actions.

**Problem statement**  Over multiple rounds, the agent and adversary intervene simultaneously on the SCM, with known DAG $\mathcal{G}$ and fixed but unknown functions $\boldsymbol{F} = \{f_i\}_{i=1}^m$ with $f_i : \mathcal{Z}_i \times \mathcal{A}_i \times \mathcal{A}_i' \to \mathcal{X}_i$. At round $t$ the agent selects actions $\boldsymbol{a}_{:,t} = \{a_{i,t}\}_{i=0}^m$ and obtains observations $\boldsymbol{x}_{:,t} = \{x_{i,t}\}_{i=0}^m$, where we add an additional subscript to denote the round of interaction. When obtaining observations, the agent also observes what actions the adversary chose $\boldsymbol{a}_{:,t}' = \{a_{i,t}'\}_{i=0}^m$. We assume the adversary does not have the power to know $\boldsymbol{a}_{:,t}$ when selecting $\boldsymbol{a}_{:,t}'$, but only has access to the history of interactions until round $t$ and knowledge of the agent's algorithm. The agent obtains a reward given by

$$y_t = f_m(\boldsymbol{z}_{m,t}, a_{m,t}, a_{m,t}') + \omega_{m,t}, \tag{2}$$

which implicitly depends on the whole action vector $\boldsymbol{a}_{:,t}$ and adversary actions $\boldsymbol{a}_{:,t}'$.

The agent's goal is to select a sequence of actions that maximizes their cumulative expected reward $\sum_{t=1}^T r(\boldsymbol{a}_{:,t}, \boldsymbol{a}_{:,t}')$ where $r(\boldsymbol{a}_{:,t}, \boldsymbol{a}_{:,t}') = \mathbb{E}[y_t \mid \boldsymbol{a}_{:,t}, \boldsymbol{a}_{:,t}']$ and expectations are taken over $\boldsymbol{\omega}$ unless otherwise stated. The challenge for the agent lies in not knowing a-priori neither the causal model (i.e., the functions $\boldsymbol{F} = \{f_i\}_{i=1}^m$), nor the sequence of adversarial actions $\{\boldsymbol{a}_{:,t}'\}_{t=1}^{\cdots}$.

**Performance metric**  After $T$ timesteps, we can measure the performance of the agent via the notion of regret:

$$R(T) = \max_{\boldsymbol{a} \in \mathcal{A}} \sum_{t=1}^T r(\boldsymbol{a}, \boldsymbol{a}_{:,t}') - \sum_{t=1}^T r(\boldsymbol{a}_{:,t}, \boldsymbol{a}_{:,t}'), \tag{3}$$

i.e., the difference between the best cumulative expected reward obtainable by playing a single fixed action if the adversary's action sequence and $\boldsymbol{F}$ were known in hindsight, and the agent's cumulative expected reward. We seek to design algorithms for the agent that are *no-regret*, meaning that $R(T)/T \to 0$ as $T \to \infty$, for any sequence $\boldsymbol{a}_{:,t}'$. We emphasize that while we use the term 'adversary', our regret notion encompasses all strategies that the adversary could use to select actions. This might include cooperative agents or mechanism non-stationarities.

For simplicity, we consider only adversary actions observed after the agent chooses actions. Our methods can be extended to also consider adversary actions observed *before* the agent chooses actions, i.e., a *context*. This results in learning a policy that returns actions depending on the context, rather than just learning a fixed action. This extension is straightforward and we briefly discuss it in Appendix A.2.

**Regularity assumptions**  We consider standard smoothness assumptions for the unknown functions $f_i : \mathcal{S} \to \mathcal{X}_i$ defined over a compact domain $\mathcal{S}$ (Srinivas et al., 2010). In particular, for each node $i \in [m]$, we assume that $f_i(\cdot)$ belongs to a reproducing kernel Hilbert space (RKHS) $\mathcal{H}_{k_i}$, a space of smooth functions defined on $\mathcal{S} = \mathcal{Z}_i \times \mathcal{A}_i \times \mathcal{A}_i'$. This means that $f_i \in \mathcal{H}_{k_i}$ is induced by a kernel function $k_i : \mathcal{S} \times \mathcal{S} \to \mathbb{R}$. We also assume that $k_i(s, s') \leq 1$ for every $s, s' \in \mathcal{S}$ [4]. Moreover, the RKHS

---

[3]There may be constraints on the actions our agent can take. We refer the reader to Sussex et al. (2022) for how our setup can be extended to handle constraints.

[4]This is known as the bounded variance property, and it holds for many common kernels.

norm of $f_i(\cdot)$ is assumed to be bounded $\|f_i\|_{k_i} \le \mathcal{B}_i$ for some fixed constant $\mathcal{B}_i > 0$. Finally, to ensure the compactness of the domains $\mathcal{Z}_i$, we assume that the noise $\boldsymbol{\omega}$ is bounded, i.e., $\omega_i \in [-1,1]^d$.

## 3 RELATED WORK

**Causal Bayesian optimization**  Several recent works study how to perform Bayesian optimization on systems with an underlying causal graph. Aglietti et al. (2020) proposes the first CBO setting with hard interventions and an algorithm that uses the do-calculus to generalise from observational to interventional data, even in settings with unobserved confounding. Astudillo and Frazier (2021) consider a noiseless setting with soft interventions (known as a function network) where a full system model is learnt, and an expected improvement objective used to select interventions. Sussex et al. (2022) propose MCBO, an algorithm with theoretical guarantees that can be used with both hard and soft interventions. MCBO propagates epistemic uncertainty about causal mechanisms through the graph, balancing exploration and exploitation using the optimism principle (Srinivas et al., 2010). Causal bandits, which similarly incorporate causal graph knowledge into the bandit setting, usually consider discrete actions with categorical observations (Lattimore et al., 2016) or linear mechanisms with continuous observations (Varici et al., 2022). All of these methods consider only stationary environments and do not account for possible adversaries.

**Bayesian optimization in non-i.i.d. settings**  Multiple works study how to develop robust strategies against shifts in uncontrollable covariates. They study notions of worst-case adversarial robustness (Bogunovic et al., 2018), distributional robustness (Kirschner et al., 2020; Nguyen et al., 2020), robust mixed strategies (Sessa et al., 2020a) and risk-aversion to uncontrollable environmental parameters (Makarova et al., 2021; Iwazaki et al., 2021). Nonstationarity is studied in the canonical BO setup in Kirschner et al. (2020); Nguyen et al. (2020) and in the CBO setup in Aglietti et al. (2021). However, these methods do not accommodate adversaries in the system, e.g., multiple agents that we cannot control. A foundation for our work is the GP-MW algorithm (Sessa et al., 2019) which studies learning in unknown multi-agents games and is a special case of our setting. We compare further with GP-MW in Section 5. Another special case of CBO-MW with a specific graph is STACKELUCB (Sessa et al., 2021), designed for playing unknown Stackelberg games with multiple types of opponent (see Appendix A.1.1).

## 4 METHOD

In this section, we introduce the methodology for the proposed CBO-MW algorithm.

### 4.1 CALIBRATED MODELS

An important component of our approach is the use of calibrated uncertainty models to learn functions $\boldsymbol{F}$, as done in Sussex et al. (2022). At the end of each round $t$, we use the dataset $\mathcal{D}_t = \{\boldsymbol{z}_{:,1:t}, \boldsymbol{a}_{:,1:t}, \boldsymbol{a}'_{:,1:t}, \boldsymbol{x}_{:,1:t}\}$ of past interventions to fit a separate model for every node in the system. CBO-MW can be applied with any set of models that have calibrated uncertainty. That is, for every node $i$ at time $t$ the model has a mean function $\mu_{i,t}$ and variance function $\sigma_{i,t}$ (learnt from $\mathcal{D}_t$) that accurately capture any epistemic uncertainty in the true model.

**Assumption 1** (*Calibrated model*)**.** All statistical models are *calibrated* w.r.t. $\boldsymbol{F}$, so that $\forall i, t$ there exists a sequence $\beta_t \in \mathbb{R}_{>0}$ such that, with probability at least $(1-\delta)$, for all $\boldsymbol{z}_i, a_i, a'_i \in \mathcal{Z}_i \times \mathcal{A}_i \times \mathcal{A}'_i$ we have $|f_i(\boldsymbol{z}_i, a_i, a'_i) - \mu_{i,t-1}(\boldsymbol{z}_i, a_i, a'_i)| \le \beta_t \sigma_{i,t-1}(\boldsymbol{z}_i, a_i, a'_i)$, element-wise.

If the models are calibrated, we can form confidence bounds that contain the true system model with high probability. This is done by combining separate confidence bounds for the mechanism at each node. At time $t$, the known set $\mathcal{M}_t$ of statistically plausible functions $\tilde{\boldsymbol{F}} = \{\tilde{f}_i\}_{i=0}^m$ is defined as:

$$
\mathcal{M}_t = \left\{ \tilde{\boldsymbol{F}} = \{\tilde{f}_i\}_{i=0}^m, \text{ s.t. } \forall i: \; \tilde{f}_i \in \mathcal{H}_{k_i}, \; \|\tilde{f}_i\|_{k_i} \le \mathcal{B}_i, \text{and} \right.
$$
$$
\left. \left| \tilde{f}_i(\boldsymbol{z}_i, a_i, a'_i) - \mu_{i,t-1}(\boldsymbol{z}_i, a_i, a'_i) \right| \le \beta_t \sigma_{i,t-1}(\boldsymbol{z}_i, a_i, a'_i), \; \forall \boldsymbol{z}_i \in \mathcal{Z}_i, a_i \in \mathcal{A}_i, a'_i \in \mathcal{A}'_i \right\}. \tag{4}
$$

**GP models** Gaussian Process (GP) models can be used to model epistemic uncertainty. These are the model class we study in our analysis (Section 5), where we also give explicit forms for $\beta_t$ that satisfy Assumption 1. For all $i \in [m]$, let $\mu_{i,0}$ and $\sigma_{i,0}^2$ denote the prior mean and variance functions for each $f_i$, respectively. Since $\boldsymbol{\omega}_i$ is bounded, it is also subgaussian and we denote the variance by $b_i^2$. The corresponding posterior GP mean and variance, denoted by $\mu_i, t$ and $\sigma_{i,t}^2$ respectively, are computed based on the previous evaluations $\mathcal{D}_t$:

$$\mu_{i,t}(\boldsymbol{s}_{i,1:t}) = \mathbf{k}_t(\boldsymbol{s}_{i,1:t})^\top (\mathbf{K}_t + b_i^2 \mathbf{I}_t)^{-1} \boldsymbol{x}_{i,1:t} \tag{5}$$

$$\sigma_{i,t}^2(\boldsymbol{s}_{i,1:t}) = k(\boldsymbol{s}_{i,1:t}; \boldsymbol{s}_{i,1:t}) - \mathbf{k}_t(\boldsymbol{s}_{i,1:t})^\top (\mathbf{K}_t + b_i^2 \mathbf{I}_t)^{-1} \mathbf{k}_t(\boldsymbol{s}_{i,1:t}), \tag{6}$$

where $\boldsymbol{s}_{i,1:t} = (\boldsymbol{z}_{i,1:t}, \boldsymbol{a}_{i,1:t}, \boldsymbol{a}'_{i,1:t})$, $\mathbf{k}_t(\boldsymbol{s}_{i,1:t}) = [k(\boldsymbol{s}_{i,j}, \boldsymbol{s}_{i,1:t})]_{j=1}^t$, and $\mathbf{K}_t = [k(\boldsymbol{s}_{i,j}, \boldsymbol{s}_{i,j'})]_{j,j'}$ is the kernel matrix.

## 4.2 THE CBO-MW ALGORITHM

We can now present the proposed CBO-MW algorithm. Our approach is based upon the classic multiplicative weights method (Littlestone and Warmuth, 1994; Freund and Schapire, 1997), widely used in adversarial online learning. Indeed, ACBO can be seen as a specific structured online learning problem. At time $t$, CBO-MW maintains a weight $w_{\boldsymbol{a}}^t$ for every possible intervention $\boldsymbol{a} \in \mathcal{A}$ such that $\sum_{\boldsymbol{a}} w_{\boldsymbol{a}}^t = 1$ and uses these to sample the chosen intervention, i.e., $\boldsymbol{a}_t \sim w_{\boldsymbol{a}}^t$. Contrary to standard CBO (where algorithms can choose actions deterministically), in adversarial environments such as ACBO randomization is necessary to achieve no-regret, see, e.g., Cesa-Bianchi and Lugosi (2006).

---

**Algorithm 1** Causal Bayesian Optimization Multiplicative Weights (CBO-MW)

---

**Require:** parameters $\tau, \{\beta_t\}_{t \geq 1}, \mathcal{G}, \boldsymbol{\Omega}$, kernel functions $k_i$ and prior means $\mu_{i,0} = 0 \ \forall i \in [m]$

1: Initialize $\boldsymbol{w}^1 = \frac{1}{|\mathcal{A}|}(1, \ldots, 1) \in \mathbb{R}^{|\mathcal{A}|}$
2: **for** $t = 1, 2, \ldots$ **do**
3:     Sample $\boldsymbol{a}_t \sim \boldsymbol{w}^t$
4:     Observe samples $\{\boldsymbol{z}_{i,t}, x_{i,t}, a'_{i,t}\}_{i=0}^m$
5:     Update posteriors $\{\mu_{i,t}(\cdot), \sigma_{i,t}^2(\cdot)\}_{i=0}^m$
6:     **for** $\boldsymbol{a} \in \mathcal{A}$ **do**
7:         Compute $\mathrm{UCB}_t^{\mathcal{G}}(\boldsymbol{a}, \boldsymbol{a}'_t)$ using Algorithm 2
8:         $\hat{y}_{\boldsymbol{a}}^t = \min(1, \mathrm{UCB}_t^{\mathcal{G}}(\boldsymbol{a}, \boldsymbol{a}'_t))$
9:         $w_{\boldsymbol{a}}^{t+1} \propto w_{\boldsymbol{a}}^t \exp(\tau \cdot \hat{y}_{\boldsymbol{a}}^t)$
10:     **end for**
11: **end for**

---

Then, CBO-MW updates the weights at the end of each round based upon what action the adversary took $\boldsymbol{a}'_t$ and the observations $\boldsymbol{x}_t$. If the mechanism between actions, adversary actions, and rewards were to be completely known (i.e., the function $r(\cdot)$ in our setup), a standard MW strategy suggests updating the weight for every $\boldsymbol{a}$ according to

$$w_{\boldsymbol{a}}^{t+1} \propto w_{\boldsymbol{a}}^t \exp\left(\tau \cdot r(\boldsymbol{a}, \boldsymbol{a}'_t)\right) ,$$

where $\tau$ is a tunable learning rate. In particular, $r(\boldsymbol{a}, \boldsymbol{a}'_t)$ is the counterfactual of what would have happened, in expectation over noise, had $\boldsymbol{a}'_t$ remained fixed but the algorithm selected $\boldsymbol{a}$ instead of $\boldsymbol{a}_t$.

However, in ACBO the system is unknown and thus such counterfactual information is not readily available. On the other hand, as outlined in the previous section, we can build and update calibrated models around the unknown mechanisms and then estimate counterfactual quantities from them. Specifically, CBO-MW utilizes the confidence set $\mathcal{M}_t$ to compute an *optimistic* estimate of $r(\boldsymbol{a}, \boldsymbol{a}'_t)$:

$$\mathrm{UCB}_t^{\mathcal{G}}(\boldsymbol{a}, \boldsymbol{a}') = \max_{\tilde{\boldsymbol{F}} \in \mathcal{M}_t} \mathbb{E}_{\boldsymbol{\omega}}\left[y \mid \tilde{\boldsymbol{F}}, \boldsymbol{a}, \boldsymbol{a}'_t\right]. \tag{7}$$

Given action $\boldsymbol{a}$, opponent action $\boldsymbol{a}'_t$ and confidence set $\mathcal{M}_t$, $\mathrm{UCB}_t^{\mathcal{G}}(\boldsymbol{a}, \boldsymbol{a}')$ represents the highest expected return among all system models in this confidence set. CBO-MW uses such estimates to update the weights in place of the true but unknown counterfactuals $r(\boldsymbol{a}, \boldsymbol{a}'_t)$. Computing $\mathrm{UCB}_t^{\mathcal{G}}(\boldsymbol{a}, \boldsymbol{a}')$ is challenging since our confidence set $\mathcal{M}_t$ consists of a set of $m$ different models and one must propagate epistemic uncertainty through all models in the system, from actions to rewards. Because mechanisms can be non-monotonic and nonlinear, one cannot simply independently maximize the output of every mechanism. We thus defer this task to an algorithmic subroutine (denoted causal UCB oracle) which we describe in the next subsection. CBO-MW is summarized in Algorithm 1.

We note that CBO-MW strictly generalizes the GP-MW algorithm of Sessa et al. (2019), which was first to propose combining MW with optimistic counterfactual reward estimates. However, they consider a trivial causal graph with only a target node, thus a *single* GP model. For this simpler

model one can compute $\text{UCB}_t^{\mathcal{G}}$ in closed-form but must ignore any causal structure in the reward model. In Section 5 and in our experiments we show CBO-MW can significantly outperform GP-MW both theoretically and experimentally.

## 4.3 CAUSAL UCB ORACLE

The problem in Eq. (7) is not amenable to commonly used optimization techniques, due to the maximization over a set of functions with bounded RKHS norm. Therefore, similar to Sussex et al. (2022) we make use of the reparameterization trick to write any $\tilde{f}_i \in \tilde{\boldsymbol{F}} \in \mathcal{M}_t$ using a function $\eta_i : \mathcal{Z}_i \times \mathcal{A}_i \times \mathcal{A}_i' \to [-1, 1]$ as

$$\tilde{f}_{i,t}(\tilde{\boldsymbol{z}}_i, \tilde{a}_i, \tilde{a}_i') = \mu_{i,t-1}(\tilde{\boldsymbol{z}}_i, \tilde{a}_i, \tilde{a}_i') + \beta_t \sigma_{i,t-1}(\tilde{\boldsymbol{z}}_i, \tilde{a}_i, \tilde{a}_i') \eta_i(\tilde{\boldsymbol{z}}_i, \tilde{a}_i, \tilde{a}_i'), \tag{8}$$

where $\tilde{x}_i = \tilde{f}_i(\tilde{z}_i, \tilde{a}_i, \tilde{a}_i') + \tilde{\omega}_i$ denotes observations from simulating actions in one of the plausible models, and not necessarily the true model. The validity of this reparameterization comes directly from the definition of $\mathcal{M}_t$ in Eq. (4) and the range of $\eta_i$.

The reparametrization allows for rewriting $\text{UCB}_t^{\mathcal{G}}$ in terms of $\boldsymbol{\eta} : \mathcal{Z} \times \mathcal{A} \times \mathcal{A}' \to [-1, 1]^{|\mathcal{X}|}$:

$$\text{UCB}_t^{\mathcal{G}}(\boldsymbol{a}, \boldsymbol{a}') = \max_{\boldsymbol{\eta}(\cdot)} \mathbb{E}_{\boldsymbol{\omega}} \left[ y \mid \tilde{\boldsymbol{F}}, \boldsymbol{a}, \boldsymbol{a}_t' \right], \text{ s.t. } \tilde{\boldsymbol{F}} = \{\tilde{f}_{i,t}\} \text{ in Eq. (8).} \tag{9}$$

In practice, we can parameterize $\boldsymbol{\eta}$, for example with neural networks, and maximize this objective using stochastic gradient ascent, as described in Algorithm 2. The use of the reparameterization trick simplifies the optimization problem because we go from optimizing over functions with a tricky constraint ($\tilde{\boldsymbol{F}} \in \mathcal{M}_t$) to a much simpler constraint ($\boldsymbol{\eta}$ just needing output in $[0, 1]$). Since the optimization problem is still non-convex, we deploy multiple random re-initializations of the $\boldsymbol{\eta}$ parameters.

---

**Algorithm 2** Causal UCB Oracle

---

**Require:** neural networks $\boldsymbol{\eta}$, actions $\boldsymbol{a}, \boldsymbol{a}'$, model posteriors $\boldsymbol{\mu}, \boldsymbol{\sigma}$, parameter $\beta_t$, repeats $N_{\text{rep}}$.
1: Initialize SOLUTIONS $= \emptyset$
2: **for** $j = 1, \ldots, N_{\text{rep}}$ **do**
3:      Randomly initialize weights of each $\eta_i \in \boldsymbol{\eta}$
4:      $\text{UCB}_{t,j}^{\mathcal{G}} = \max_{\boldsymbol{\eta}(\cdot)} \mathbb{E}[y \mid \tilde{\boldsymbol{F}}, \boldsymbol{a}, \boldsymbol{a}']$ computed via stochastic gradient ascent on $\boldsymbol{\eta}$
5:      SOLUTIONS $=$ SOLUTIONS $\cup \{\text{UCB}_{t,j}^{\mathcal{G}}\}$.
6: **end for**
7: **return** $\max(\text{SOLUTIONS})$

---

## 5 ANALYSIS

Here we analyse the theoretical performance of CBO-MW and provide a bound on its regret as a function of the underlying causal graph. For our analysis, we make some additional technical assumptions. First, we assume all $f_i \in \boldsymbol{F}$ are $L_f$-Lipschitz continuous. This follows directly from the regularity assumptions of Section 2. Second, we assume that $\forall i, t$, the functions $\mu_i, \sigma_{i,t}$ are $L_\mu$, $L_\sigma$ Lipschitz continuous. This holds if the RKHS of each $f_i$ has a Lipschitz continuous kernel (see Curi et al. (2020), Appendix G). Finally, we assume the causal UCB oracle can always compute $UCB_t^G(a, a')$ (Eq. (9)) exactly.

In addition, to show how the regret guarantee depends on the specific GP hyperparameters used, we use a notion of model complexity for each node $i$:

$$\gamma_{i,T} := \max_{A_i \subset \{\mathcal{Z}_i \times \mathcal{A}_i \times \mathcal{A}_i'\}^T} I(\boldsymbol{x}_{i,1:T}, f_i) \tag{10}$$

where $I$ is the mutual information and the observations $\boldsymbol{x}_{i,1:T}$ implicitly depend on the GP inputs $A_i$. This is analogous to the maximum information gain used in the analysis of standard BO algorithms Srinivas et al. (2010). We also define

$$\gamma_T = \max_i \gamma_{i,T} \tag{11}$$

as the worst-case maximum information gain across all nodes in the system.

Finally, we define two properties of the causal graph structure that the regret guarantee will depend on. In the DAG $\mathcal{G}$ over nodes $\{X_i\}_{i=0}^m$, let $\Delta$ denote the maximum number of parents of any variable in $\mathcal{G}$: $\Delta = \max_i |pa(i)|$. Then let $N$ denote the maximum distance from a root node to $X_m$: $N = \max_i \text{dist}(X_i, X_m)$ where $\text{dist}(\cdot, \cdot)$ is the number of edges in the longest path from a node $X_i$ to $X_m$. We can then prove the following theorem on the regret of CBO-MW.

**Theorem 1.** *Fix* $\delta \in (0, 1)$, *if actions are played according to* CBO-MW *with* $\beta_t = \mathcal{O}\left(\mathcal{B} + \sqrt{\gamma_{t-1} + \log(m/\delta)}\right)$ *and* $\tau = \sqrt{(8 \log |\mathcal{A}|)/T}$, *then with probability at least* $1 - \delta$,

$$R(T) = \mathcal{O}\left(\sqrt{T \log |\mathcal{A}|} + \sqrt{T \log(2/\delta)} + \left(\mathcal{B} + \sqrt{\gamma_T + \log(m/\delta)}\right)^{N+1} \Delta^N L_\sigma^N L_f^N m \sqrt{T \gamma_T}\right),$$

*where* $\mathcal{B} = \max_i \mathcal{B}_i$, $\gamma_T = \max \gamma_{i,t}$. *That is,* $\gamma_T$ *is the worst-case maximum information gain of any of the GP models.*

Theorem 1, whose proof is deferred to the appendix, shows that CBO-MW is no-regret for a variety of common kernel functions, for example linear and squared exponential kernels. This is because even when accounting for the dependence of $\gamma_T$ in $T$, the bound is still sublinear in $T$. We discuss the dependence of $\gamma_T$ on $T$ for specific kernels in Appendix A.1.2.

**Comparison with GP-MW**    We can use Theorem 1 to demonstrate that the use of graph structure in CBO-MW results in a potentially *exponential* improvement in the rate of regret compared to GP-MW (Sessa et al., 2019) with respect to the number of action variables $m$. Consider the graph in Fig. 4b (see Appendix) for illustration. When all $X_i$ in CBO-MW are modeled with squared exponential kernels, we have $\gamma_T = \mathcal{O}((\Delta + 2)(\log T)^{\Delta+3})$. This results in a cumulative regret that is exponential in $\Delta$ and $N$. Instead, GP-MW uses a single high-dimensional GP (Fig. 4a), implying $\gamma_T = \mathcal{O}((\log T)^m)$ for a squared exponential kernel. Note that $m \geq \Delta + N$ and thus, for several common graphs, the exponential scaling in $N$ and $\Delta$ could be significantly more favorable than the exponential scaling in $m$. Specifically for the binary tree of Fig. 4b, where $N = \log(m)$ and $\Delta = 2$, the cumulative regret of CBO-MW will have only *polynomial* dependence on $m$.

Furthermore, in addition to favorable scaling of the regret in $m$, the model class considered by CBO-MW is considerably larger than that of GP-MW, because CBO-MW can model systems where reward depends on actions according to a composition of GPs based on $\mathcal{G}$, rather than a single GP.

## 6    Computational Considerations in Larger Action Spaces

The computational complexity of CBO-MW is dominated by the computation of the counterfactual expected reward estimate for each possible intervention $\boldsymbol{a} \in \mathcal{A}$ (Line 7 in Algorithm 1). In many situations, even with a large number of action variables $m$, $|\mathcal{A}|$ may still be small and thus CBO-MW feasible to implement. A notable example is when there exist constraints on the possible interventions, such as only being able to intervene on at most a few nodes simultaneously. In the worst case, though, $|\mathcal{A}|$ grows exponentially with $m$ and thus CBO-MW may not be computationally feasible. In this section we will show how prior knowledge of the problem structure can be used to modify CBO-MW to be computationally efficient even in settings with huge action spaces. We note that the computational efficiency of CBO-MW is not affected by the number of possible adversary interventions $|\mathcal{A}'|$ because these are only observed a-posteriori (in fact, $\mathcal{A}'$ need not be finite).

The general idea consists of decomposing CBO-MW into a *decentralized* algorithm, which we call D-CBO-MW, that orchestrates multiple sub-agents. First recall that $\mathcal{A}$ can be decomposed into $m$ smaller action spaces so that $\mathcal{A} = \mathcal{A}_1 \times ... \times \mathcal{A}_m$. We then consider $m$ independent agents where each agent $i$ performs CBO-MW but only over the action space $\mathcal{A}_i$. Importantly, the observations of the actions of the other agents are considered part of $\mathcal{A}'$ for that agent. Moreover, all agents utilize the same calibrated model $\mathcal{M}_t$ at each round. We provide full pseudo-code and theory in the appendix. Our approach is inspired by Sessa et al. (2021) who propose a distributed analog of the GP-MW algorithm.

In Appendix B we show that under specific assumptions on the reward function, D-CBO-MW provably enjoys an approximate version of the guarantees in Theorem 1. Namely, we study a setting where $r$ is a *submodular* and monotone increasing function of $a$ for any given set of adversary actions. Submodularity is a diminishing returns property (see formal definition in the appendix) widely exploited in a variety of domains to derive efficient methods with approximation guarantees, see e.g., Marden and Wierman (2013); Sessa et al. (2021); Paccagnan and Marden (2022). Similarly, we exploit submodularity in the overall reward's structure to parallelize the computation over the possible interventions in our causal graph. In our experiments, we study rebalancing an SMS where CBO-MW is not applicable due to a combinatorial action space but D-CBO-MW is efficient and achieves good performance.

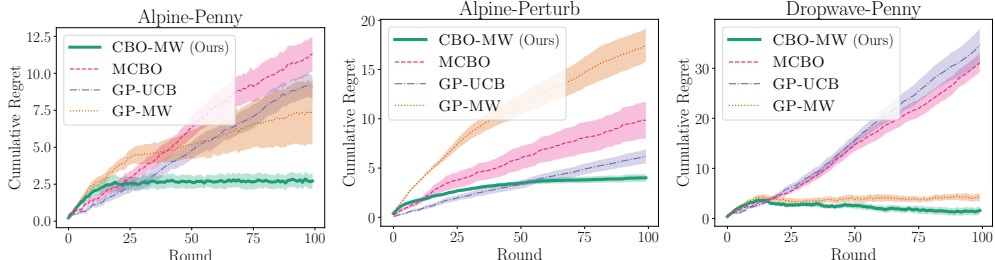

Figure 2: CBO-MW achieves sublinear regret and high sample efficiency on 3 function networks in the presence of adversaries. Non-adversarial methods (GP-UCB and MCBO) achieve linear regret and non-causal methods (GP-MW) are less sample efficient.

# 7 EXPERIMENTS

We evaluate CBO-MW and D-CBO-MW on various synthetic problems and a simulator of rebalancing an SMS based on real data. The goal of the experiments is to understand how the use of causal modelling and adaptability to external factors in CBO-MW affects performance compared to other BO methods that are missing one or both of these components. All methods are evaluated over 10 repeats, with mean and standard error reported for different time horizons.

## 7.1 FUNCTION NETWORKS

**Networks** We evaluate CBO-MW on 8 diverse environments. As a base, we take 4 examples of function networks from Astudillo and Frazier (2021). Function networks is a noiseless CBO setup with no adversaries. To study an adversarial setup, we modify each environment by adding adversaries' inputs in 2 ways: Penny and Perturb. In Penny, an adversary can affect a key node with a dynamic similar to the classic matching pennies game. In Perturb, the adversary can perturb some of the agent's interventions. The exact way in which the adversary's actions affect the environment is unknown and the actions themselves can only be observed a-posteriori. In each round, the adversary has a 20% chance to play randomly, and an 80% chance to try and minimize the agent's reward, using full knowledge of the agent's strategy and the environment. The causal graph and structural equation model for each environment in given in Appendix C.

**Baselines** We compare the performance of CBO-MW (Algorithm 1) with GP-MW Sessa et al. (2019) which does not exploit the causal structure. We additionally compare against non-adversarial baselines GP-UCB (Srinivas et al., 2010), and MCBO (Sussex et al., 2022) which uses a causal model but cannot account for adversaries.

**Results** We give results from 3 of the 8 environments in Fig. 2. Others can be found in Appendix C. In general across the 8 environments, MCBO and GP-UCB obtain linear regret, while the regret of GP-MW and CBO-MW grows sublinearly, consistent with Theorem 1. CBO-MW has the strongest or joint-strongest performance on 7 out of the 8 environments. The setting where CBO-MW is not strongest involves a dense graph where worst-case GP sparsity is the same as in GP-MW, which is consistent with our theory. In settings such as Alpine where the graph is highly sparse, we observe the greatest improvements from using CBO-MW.

## 7.2 LEARNING TO REBALANCE SHARED MOBILITY SYSTEMS (SMSS)

We evaluate D-CBO-MW on the task of rebalancing an SMS, a setting introduced in Sessa et al. (2021). The goal is to allocate bikes to city locations (using relocation trucks, overnight) to maximize the number of bike trips in the subsequent day. The action space is combinatorial because each of the 5 trucks can independently reallocate units to a new depot. This makes the approaches studied in Section 7.1 computationally infeasible, so we deploy D-CBO-MW. We expect the reward $Y_t$ (total trips given unit allocation) to be monotone submodular because adding an extra unit to a depot should increase total trips but with decreasing marginal benefit, as discussed in Sessa et al. (2021). The goal is to compare D-CBO-MW to a distributed version of GP-MW and understand whether the use of a causal graph can improve sample efficiency.

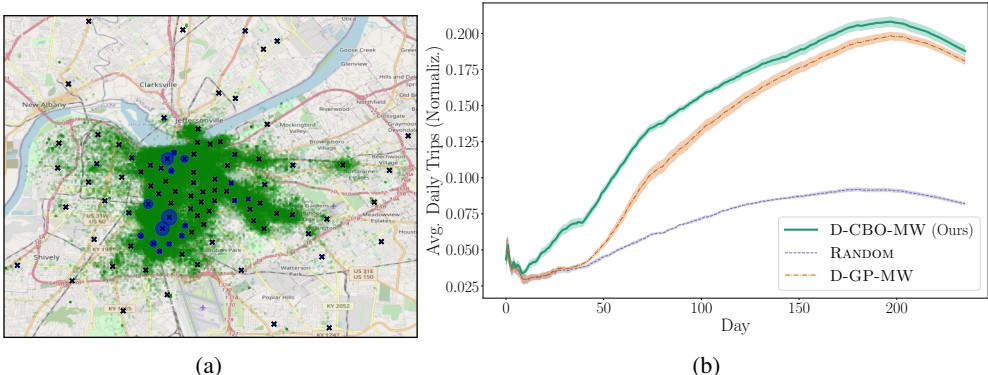

| (a) | (b) |

Figure 3: Learning to rebalance an SMS. (**a**) Green dots represent demand for trips, black crosses are depot locations, and blue circles are scaled in size to the frequency that D-CBO-MW assigns bikes to that depot (averaged across all repeats and days). D-CBO-MW assigns bikes in strategic locations near high demand. (**b**) D-CBO-MW fulfills more total trips across over 200 days compared to D-GP-MW which does not use a causal reward model.

**Setup and trips simulator**  A simulator is constructed using historical data from an SMS in Louisville, KY (Lou, 2021). Before each day $t$, all 40 bikes in the system are redistributed across 116 depots. This is done by 5 trucks, each of which can redistribute 8 bikes at one depot, meaning a truck $i$'s action is $a_i \in [116]$. The demand for trips corresponds to real trip data from Lou (2021). After each day, we observe weather and demand data from the previous day which are highly non-stationary and thus correspond to adversarial interventions $a'$ according to our model. Our simulator is identical to the one of Sessa et al. (2021), except we exclude weekends for simplicity. We give more details in Appendix C.

We compare three methods on the SMS simulator. First, in RANDOM each truck places its bikes at a depot chosen uniformly at random. Second, D-GP-MW modifies GP-MW using the same ideas as those presented in Section 6. It is a special case of D-CBO-MW but using the graph in Fig. 1(a). That is, a single GP is used to predict $Y_t$ given $a_t, a'_t$. Finally, we evaluate D-CBO-MW which utilizes a more structured causal graph exploiting the reward structure. Based on historical data, we cluster the depots into regions such that trips don't frequently occur across two different regions (e.g., when such regions are too far away). Then, our graph models the trips starting in each region only using bike allocations to depots in that region. $Y_t$ is computed by summing the trips across all regions (see Fig. 8 (b) in the appendix for an illustration). This system model uses many low-dimensional GPs instead of a single high-dimensional GP as used in D-GP-MW.

**Results**  Fig. 3 (a) displays the allocation strategy of D-CBO-MW. We observe that D-CBO-MW learns the underlying demand patterns and allocates bikes in strategic areas where the demand (green dots) is higher. This plot is zoomed-in for readability. The allocation strategy over all depots is shown in Appendix Fig. 9. Moreover, in Fig. 3 (b) we see that D-CBO-MW is significantly more sample efficient than D-GP-MW. This improvement is largely attributed to early rounds, where D-CBO-MW learns the demand patterns much faster than D-GP-MW due to learning smaller-dimensional GPs.

## 8 CONCLUSION

We introduce CBO-MW, the first principled approach to causal Bayesian optimization in non-stationary and potentially multi-agent environments. We prove a sublinear regret guarantee for CBO-MW and demonstrate a potentially exponential improvement in regret, in terms of the number of possible intervention targets, compared to state-of-the-art methods. We further propose a distributed version of our approach, D-CBO-MW, that can scale to large action spaces and achieves approximate regret guarantees when rewards are monotone submodular. Empirically, our algorithms outperform existing non-causal and non-adversarial methods on synthetic function network tasks and on an SMS rebalancing simulator based on real data.

## REPRODUCIBILITY STATEMENT

Attached to this submission we include code (`https://github.com/ssethz/acbo`) that implements CBO-MW, D-CBO-MW and all baselines seen in the experiments. It includes code for all function network environments and the SMS setting, which are also detailed in the appendix. All datasets used in the SMS setting are also included at this url. The appendix includes information on our experimental protocol, for example how we selected the hyperparameters for the experiments shown in the paper.

All technical assumptions are given in the main paper, and complete proofs of all theoretical results are given in the appendix. Pseudocode for CBO-MW is given in the main paper and pseudocode for D-CBO-MW is given in the appendix.

## ACKNOWLEDGEMENTS

We thank Lars Lorch for his feedback on the draft of this paper.

This research was supported by the Swiss National Science Foundation under NCCR Automation, grant agreement 51NF40 180545, by the European Research Council (ERC) under the European Union's Horizon grant 815943, and by ELSA (European Lighthouse on Secure and Safe AI) funded by the European Union under grant agreement No. 101070617.

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

ADVERSARIAL CAUSAL BAYESIAN OPTIMIZATION

# A APPENDIX

## A.1 PROOF OF THEOREM 1

We start by proving that there exists a sequence of $\beta_t$ that allow Assumption 1 to hold.

**Lemma 1** (Node Confidence Lemma ). *Let $\mathcal{H}_{k_i}$ be a RKHS with underlying kernel function $k_i$. Consider an unknown function $f_i : \mathcal{Z}_i \times \mathcal{A}'_i \times \mathcal{A}'_i \to \mathcal{X}_i$ in $\mathcal{H}_{k_i}$ such that $\|f\|_{k_i} \leq \mathcal{B}_i$, and the sampling model $x_{i,t} = f(z_{i,t}, a_{i,t}, a'_{i,t}) + \omega_t$ where $\omega_t$ is b-sub-Gaussian (with independence between times). By setting*

$$\beta_t = \mathcal{B}_i + b\sqrt{2\left(\gamma_{t-1} + \log(m/\delta)\right)},$$

*the following holds with probability at least $1 - \delta$:*

$$|\mu_{t-1}(z_i, a_i, ai') - f_i(z_i, a_i, a'_i)| \leq \beta_t \sigma_{i,t-1}(z_i, a_i, a'_i),$$

*$\forall z_i, a_i, a'_i \in \mathcal{Z}'_i \times \mathcal{A}_i \times \mathcal{A}'_i, \quad \forall t \geq 1, \quad \forall i \in [m], \quad$ where $\mu_{t-1}(\cdot)$ and $\sigma_{t-1}(\cdot)$ are given in Eq. (5), Eq. (6) and $\gamma_{t-1}$ is the maximum information gain defined in (11).*

*Proof.* Lemma 1 follows directly from Sessa et al. (2019, Lemma 1) after applying a union bound so that the statement holds for all $m$ GP models in our system. □

Now we give an upper and lower confidence bound for $r$ at each $t$. Note that since in our setup all $\omega_i$ are assumed bounded in $[0, 1]$, we can use $b = \frac{1}{4}$.

**Lemma 2** (Reward Confidence Lemma). *Choose $\delta \in [0, 1]$ and then choose the sequence $\{\beta_t\}_{t=1}^T$ according to Lemma 1.*

*$\forall a, a_i \in \mathcal{A} \times \mathcal{A}'$ , $\forall t$ we have with probability $1 - \delta$*

$$\text{UCB}_t^{\mathcal{G}}(a, a') - C_t(\delta) \leq r(a, a') \leq \text{UCB}_t^{\mathcal{G}}(a, a'_t)$$

*where we define $C_t(\delta)$ as*

$$C_t(\delta) = L_{Y,t} \Delta^N \sqrt{m \mathbb{E}_{\boldsymbol{\omega}}\left[\sum_{i=0}^m \left\|\sigma_{i,t-1}(z_{i,t}, a_{i,t}, a'_{i,t})\right\|^2\right]}.$$

*We define $L_{Y,t} = 2\beta_t(1 + L_f + 2\beta_t L_\sigma)^N$.*

*Proof.* The RHS follows firstly from Assumption 1 meaning that with probability at least $1 - \delta$, $f \in \{\tilde{f}_t\}$. The RHS then follows directly from the definition of $\text{UCB}^{\mathcal{G}}$ in Eq. (7).

The LHS follows directly from Sussex et al. (2022, Lemma 4). The addition of the adversary actions does not change the analysis in this lemma because for a given $t$, $a'_t$ is fixed for both the true model $f$ and the model in $\{\tilde{f}\}$ that leads to the upper confidence bound. □

Now we can prove the main theorem. Choose $\delta \in [0, 1]$ and then choose the sequence $\{\beta_t\}_{t=1}^T$ according to Lemma 1 so that Assumption 1 holds with probability $1 - \frac{\delta}{2}$. First recall that regret is defined as

$$R(T) = \sum_{t=1}^T r(\bar{a}, a'_t) - \sum_{t=1}^T r(a_t, a'_t)$$

where $\bar{a} = \arg\max \sum_{t=1}^T r(\bar{a}, a'_t)$. Now we can say that with probability at least $1 - \frac{\delta}{2}$

$$R(T) \leq \sum_{t=1}^{T} \min\{1, \mathrm{UCB}_t^{\mathcal{G}}(\bar{\boldsymbol{a}}, \boldsymbol{a}_t')\} - \sum_{t=1}^{T} \left[ \mathrm{UCB}_t^{\mathcal{G}}(\boldsymbol{a}_t, \boldsymbol{a}_t') - C_t\left(\delta/2\right) \right] \tag{12}$$

$$\leq \sum_{t=1}^{T} \min\{1, \mathrm{UCB}_t^{\mathcal{G}}(\bar{\boldsymbol{a}}, \boldsymbol{a}_t')\} - \sum_{t=1}^{T} \mathrm{UCB}_t^{\mathcal{G}}(\boldsymbol{a}_t, \boldsymbol{a}_t') + \sum_{t=1}^{T} C_t\left(\delta/2\right) \tag{13}$$

where the first line follows from Lemma 2.

Evaluating the last term we see

$$\sum_{t=1}^{T} C_t\left(\delta/2\right) \overset{\text{①}}{=} \sqrt{T} \sqrt{\sum_{t=1}^{T} C_t\left(\delta/2\right)^2} \tag{14}$$

$$\overset{\text{②}}{\leq} \sqrt{T} \sqrt{\sum_{t=1}^{T} L_{Y,t}^2 \Delta^{2N} \mathbb{E}_{\boldsymbol{\omega}}\left[ \sum_{i=0}^{m} \left\| \sigma_{i,t-1}(z_{i,t}, a_{i,t}, a_{i,t}') \right\|^2 \right]} \tag{15}$$

$$\overset{\text{③}}{=} \mathcal{O}\left( L_{Y,T} \Delta^N \sqrt{Tm\gamma_T} \right). \tag{16}$$

① comes from AM-QM inequality and ② comes from plugging in for $C_t(\frac{\delta}{2})$. Finally, ③ comes from $L_{T,t}$ being weakly increasing in $t$ and from using the same analysis as (Sussex et al., 2022, Theorem 1) to upper bound the sum of $\sigma$s with $\gamma_T$.

Moreover, we can upper bound the first two terms of Eq. (13) using a standard regret bound for the MW update rule (Line 9 in CBO-MW), e.g. from (Cesa-Bianchi and Lugosi, 2006, Corollary 4.2). Indeed, with probability $1 - \frac{\delta}{2}$ it holds:

$$\sum_{t=1}^{T} \min\{1, \mathrm{UCB}^{\mathcal{G}}(\bar{\boldsymbol{a}}, \boldsymbol{a}_t')\} - \sum_{t=1}^{T} \mathrm{UCB}^{\mathcal{G}}(\boldsymbol{a}_t, \boldsymbol{a}_t') = \mathcal{O}\left( \sqrt{T \log |\mathcal{A}|} + \sqrt{T \log(2/\delta)} \right). \tag{17}$$

Using the union bound on the two different probabilistic events discussed so far (Assumption 1 and Eq. (17)) we can say that with probability at least $1 - \delta$

$$R(T) = \mathcal{O}\left( \sqrt{T \log |\mathcal{A}|} + \sqrt{T \log(2/\delta)} + L_{Y,T} \Delta^N \sqrt{Tm\gamma_T} \right).$$

Substituting in for $L_{Y,t}$ and $\beta_t$ gives the result of Theorem 1.

### A.1.1 STACKELUCB AS A SPECIAL CASE

In Fig. 4 and Section 5 we give a comparison between GP-MW and CBO-MW, and see that GP-MW can be seen as a special case of CBO-MW with a one-node causal graph. STACKELUCB Sessa et al. (2020b) is another online learning algorithm that can be seen as a special case of CBO-MW. We show the graph in Fig. 5. In STACKELUCB an agent plays an unknown Stackelberg game against a changing opponent which at time $t$ has representation $a_{0,t}'$. After the agent selects an action $a_{0,t}$, the opponent sees this action and responds based upon a fixed but unknown response mechanism for that opponent: their response is $X_{0,t} = f_0(a_{0,t}', a_{0,t})$. Then, the response, game outcome $Y_t$, and opponent identity are revealed to the agent. The sequence of opponent identities, i.e. $\{a_{0,t}'\}_{t=1}^T$, can be selected potentially adversarially based on knowledge of the agent's strategy.

### A.1.2 DEPENDENCE OF $\beta_T$ ON $T$ FOR PARTICULAR KERNELS

In Theorem 1, there is a dependence of the bound on $\gamma_T$. If $\gamma_T$ scales with at worst $\mathcal{O}\left(\sqrt{T}\right)$, then the overall bound will not be in $T$, resulting in CBO-MW being no-regret. $\gamma_T$ will depend on the

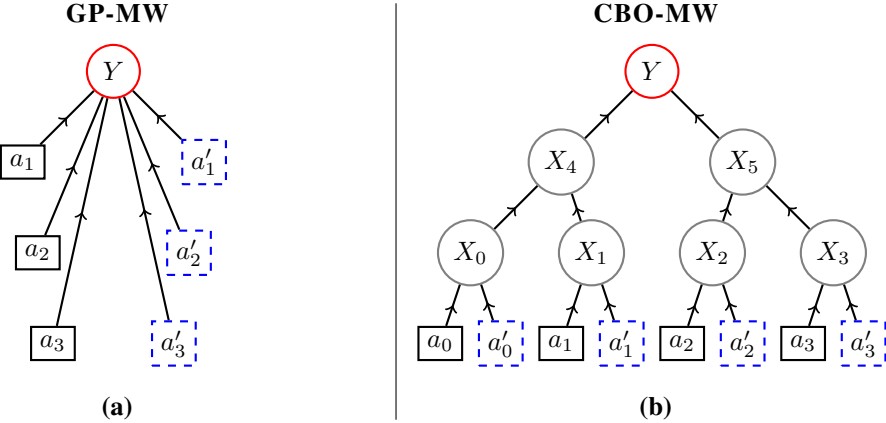

Figure 4: An example of a binary tree graph to compare GP-MW and CBO-MW. In **(a)** we see GP-MW ignores all observations and models $Y$ given all agent and adversary actions, resulting a single high-dimensional GP. Meanwhile in **(b)**, for the same task, CBO-MW fits a model for every observation resulting in multiple low-dimensional GP models.

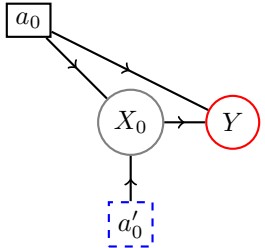

Figure 5: The causal graph corresponding to STACKELUCB, a special case of our setup. The opponent action $X_0$ is played based on the opponent type $a'_0$ and as a response to the agent's action $a_0$. The reward $Y$ depends on the agent and opponent's action.

kernel of the GP used to model each system component. For simplicity, in this section we'll assume that the same kernel is used for modelling all nodes, though if different kernels are used one just needs to consider the most complex (worst scaling of $\gamma_T$ with $T$)

For $\gamma_T$ corresponding to the linear kernel and squared exponential kernel, we have sublinear regret regardless of $N$ because $\gamma_T$ will be only logarithmic in $T$. In particular, a linear kernel leads to $\gamma_T = \mathcal{O}\left((\Delta + 1)\log T\right)$ and a squared exponential kernel leads to $\gamma_T = \mathcal{O}\left((\Delta + 1)(\log T)^{\Delta+2}\right)$.

However, for a Matérn kernel, where the best known bound is $\gamma_T = \mathcal{O}\left((T)^c log(T)\right)$ with hyperparameter $0 < c < 1$, the cumulative regret bound will not be sublinear if $N$ and $c$ are sufficiently large. A similar phenomena with the Matérn kernel appears in the guarantees of Curi et al. (2020) which use GP models in model-based reinforcement learning and in Sussex et al. (2022).

## A.2 INCORPORATING CONTEXTS INTO CBO-MW

Our approach can be easily extended to a setting where $a'_t$, the adversary actions at time $t$, are observed before the agent chooses $a_t$. In this setting the adversary actions could be thought of as contexts. Our method for computing UCB$^{\mathcal{G}}$ can be plugged into the algorithm of Sessa et al. (2020c). Their algorithm maintains a separate set of weights for every context unless two contexts are 'close' – correspond to a similar expected reward for all possible $a_t$.

---

**Algorithm 3** Multiplicative Weights Update (MWU)

---

**Require:** set $\mathcal{A}_i$ where $|\mathcal{A}_i| = K_i$, learning rate $\eta$
1: Initialize $\boldsymbol{w}^1 = \frac{1}{K_i}(1, \ldots, 1) \in \mathbb{R}^{K_i}$
2: **function** PLAY_ACTION
3:     $\boldsymbol{p} = \boldsymbol{w} \cdot 1 / \left( \sum_{j=1}^{K_i} \boldsymbol{w}[j] \right)$
4:     $a \sim \mathbf{p}$
5:     **return** a                         // sample action
6: **end function**
7:
8: **function** UPDATE$(f(\cdot))$
9:     $\boldsymbol{f} = \min(\mathbf{1}, [f(a)]_{a \in \mathcal{A}_i}) \in \mathbb{R}^K$        // rewards vector
10:     $\mathbf{w} = \mathbf{w} \cdot \exp(\eta \mathbf{f})$                      // MW update
11:     **return**
12: **end function**

---

**Algorithm 4** Distributed Causal Bayesian Optimization Multiplicative Weights (D-CBO-MW)

---

**Require:** parameters $\tau, \{\beta_t\}_{t \geq 1}, \mathcal{G}, \boldsymbol{\Omega}$, kernel functions $k_i$ and prior means $\mu_{i,0} = 0, \forall i \in [m]$
1: $\text{Algo}^i \leftarrow \text{MWU}(\mathcal{A}^i), \ i = 1, \ldots, m$, Algorithm 3        // initialize agents
2: **for** $t = 1, 2, \ldots$ **do**
3:     $\text{Algo}_i$.PLAY_ACTION$(), \ i = 1, \ldots, m$    // sample actions and perform interventions
4:     Observe samples $\{\boldsymbol{z}_{i,t}, x_{i,t}, a'_{i,t}\}_{i=0}^m$
5:     Update posteriors $\{\mu_{i,t}(\cdot), \sigma_{i,t}^2(\cdot)\}_{i=0}^m$
6:     **for** $i \in 1, \ldots, m$ **do**
7:         **for** $a_i \in \mathcal{A}_i$ **do**
8:             Compute $\text{ucb}_{i,t}(a_i) = \text{UCB}_t^{\mathcal{G}}(a_i, \boldsymbol{a}_{-i,t}, \boldsymbol{a}'_t)$ for $a_i \in \mathcal{A}_i$ using Algorithm 2
9:         **end for**
10:         $\text{Algo}_i$.UPDATE$(\text{ucb}_{i,t}(\cdot)), i = 1, \ldots, m$
11:     **end for**
12: **end for**

---

# B AN EFFICIENT VARIANT OF CBO-MW FOR LARGE ACTION SPACES

## B.1 THE DISTRIBUTED CBO-MW (D-CBO-MW) ALGORITHM

For ease of readability in the pseudocode we pull-out some key functionality of the MW algorithm, which we put into a class called MWU described in Algorithm 3. Each agent is an instance of this class, i.e., each agent maintains its own set of weights over its actions, and updates to these instances are coordinated by D-CBO-MW as described in Algorithm 4. Conceptually it can be thought of as $m$ instances of CBO-MW, where the action spaces of other agents are absorbed into $\mathcal{A}'$. While for presenting the algorithm we always decompose $\mathcal{A}$ as $\mathcal{A}_1 \times \cdots \times \mathcal{A}_m$ and have each agent control the intervention on one node, one could in practice choose to decompose the action space in a different way. A simple example would be having agent one control $\mathcal{A}_1 \times \mathcal{A}_2$ and thus designing the intervention for both $X_1, X_2$

When computing the $\text{UCB}^{\mathcal{G}}$ for a single agent $i \in [m]$, we will find the following notation convenient. Let $a_{i,t} \in \mathcal{A}_i$ be the action chosen by agent $i$ at time $t$. $\boldsymbol{a}_{-i,t} \in \mathcal{A}_{-i} \subseteq \mathcal{A}$ are the actions chosen at time $t$ by all agents except agent $i$. Note that since the subspaces of the action space each agent controls are non-overlapping, $\mathcal{A} = \mathcal{A}_i \cup \mathcal{A}_{-i}$. When agent $i$ chooses actions in $\mathcal{A}_i$, for convenience we will from now on represent it as a vector in $\mathcal{A}$ with 0 at all indexes the agent cannot intervene. We do similarly for $\boldsymbol{a}_{-i,t}$. Then we use the notation $\text{UCB}_t^{\mathcal{G}}(a_{i,t}, \boldsymbol{a}_{-i,t}, \boldsymbol{a}'_t) = \text{UCB}_t^{\mathcal{G}}(a_{i,t} + \boldsymbol{a}_{-j,t}, \boldsymbol{a}'_t)$.

## B.2 APPROXIMATION GUARANTEES FOR MONOTONE SUBMODULAR REWARDS

In this section we show that D-CBO-MW enjoys provable approximation guarantees in case of monotone and DR-submodular rewards. Both such notions are defined below.

**Definition 1** (Monotone Function). A function $f : \mathcal{A} \subseteq \mathbb{R}^m \to \mathbb{R}$ is monotone if for $\boldsymbol{x} \leq \boldsymbol{y}$,

$$f(\boldsymbol{x}) \leq f(\boldsymbol{y}).$$

**Definition 2** (DR-Submodularity, (Bian et al., 2017)). A function $f : \mathcal{A} \subseteq \mathbb{R}^m \to \mathbb{R}$ is DR-submodular if, for all $\boldsymbol{x} \leq \boldsymbol{y} \in \mathcal{A}$, $\forall i \in [m], \forall k \geq 0$ such that $(\boldsymbol{x} + k\boldsymbol{e}_i)$ and $(\boldsymbol{y} + k\boldsymbol{e}_i) \in \mathcal{A}$,

$$f(\boldsymbol{x} + k\boldsymbol{e}_i) - f(\boldsymbol{x}) \geq f(\boldsymbol{y} + k\boldsymbol{e}_i) - f(\boldsymbol{y}).$$

DR-submodularity is a generalization of the more common notion of a submodular set function to continuous domains Bach (2019). For our analysis, in particular, we assume that for every $\boldsymbol{a}'_{:,t} \in \mathcal{A}'$, the reward function $r(\boldsymbol{a}_{:,t}, \boldsymbol{a}'_{:,t})$ is monotone DR-submodular in $\boldsymbol{a}_{:,t}$.

We consider ACBO as a game played among all the distributed agents and the adversary. Our guarantees are then based on the results of Sessa et al. (2021) and depend on the following notions of average and worst-case game curvature.

**Definition 3** (Game Curvature). Consider a sequence of adversary actions $\boldsymbol{a}'_1, \ldots, \boldsymbol{a}'_T$. We define the average and worst-case game curvature as

$$c_{avg}(\{\boldsymbol{a}'_{:,t}\}_{t=1}^T) = 1 - \inf_i \frac{\sum_{t=1}^T [\nabla r(2\boldsymbol{a}_{max}, \boldsymbol{a}'_{:,t})]_i}{\sum_{t=1}^T [\nabla r(\boldsymbol{0}, \boldsymbol{a}'_{:,t})]_i} \in [0, 1],$$

$$c_{wc}(\{\boldsymbol{a}'_{:,t}\}_{t=1}^T) = 1 - \inf_{t,i} \frac{[\nabla r(2\boldsymbol{a}_{max}, \boldsymbol{a}'_{:,t})]_i}{[\nabla r(\boldsymbol{0}, \boldsymbol{a}'_{:,t})]_i} \in [0, 1],$$

where $\boldsymbol{a}_{max} = a_{max}\boldsymbol{1}$ and $a_{max}$ is the largest intervention value a single agent can assign at any index. If $2\boldsymbol{a}_{max}$ is outside the domain of $r$, then the definition can be applied to a monotone extension of $r$ over $\mathbb{R}^m$. A definition of game curvature for non-differentiable $r$ is given in the appendix of Sessa et al. (2021).

Curvature measures how close $r(\cdot, \boldsymbol{a}'_{:,t})$ is to being linear in the agents' actions, with $c_{avg} = c_{wc} = 0$ coinciding with a completely linear function. The closer $r(\cdot, \boldsymbol{a}'_{:,t})$ is to linear, generally the easier the function is to optimize in a distributed way, because a linear function is separable in its inputs. $c_{wc}$ is the worst-case curvature of $r(\cdot, \boldsymbol{a}'_{:,t})$ across all rounds $t$, while $c_{avg}$ is the average curvature across rounds. The curvature of $r(\cdot, \boldsymbol{a}'_{:,t})$ will change with $t$ because $\boldsymbol{a}'_{:,t}$ will change across rounds.

We will without loss of generality assume that $r(\boldsymbol{0}, \boldsymbol{a}') = 0$ for all $\boldsymbol{a}'$. If this did not hold then $r(\boldsymbol{0}, \boldsymbol{a}')$ could simply be subtracted from all observations to make it true. Then, we can show the following.

**Theorem 2.** *Consider the setting of Theorem 1 but with the additional monotone submodularity and curvature assumptions made in this section. Assume $|\mathcal{A}_i| = K$ for all $i$. Then, if actions are played according to* D-CBO-MW *with $\beta_t = \mathcal{O}\left(\mathcal{B} + \sqrt{\gamma_{t-1} + \log(m/\delta)}\right)$ and $\eta = \sqrt{8\log(K)/T}$ then with probability at least $1 - \delta$,*

$$\sum_{t=1}^T r(\boldsymbol{a}_t, \boldsymbol{a}'_t) \geq \alpha \cdot \text{OPT} - m \cdot \mathcal{O}\left(\left(\mathcal{B} + \sqrt{\gamma_T + \log(m/\delta)}\right)^{N+1} \Delta^N L_\sigma^N L_f^N m\sqrt{T\gamma_T}\right) \quad (18)$$

$$- m \cdot \mathcal{O}\left(\sqrt{T\log K} + \sqrt{T\log(2m/\delta)}\right),$$

*with*

$$\alpha = \max\left\{1 - c_{avg}(\{\boldsymbol{a}'_{:,t}\}_{t=1}^T), \left(1 + c_{wc}(\{\boldsymbol{a}'_{:,t}\}_{t=1}^T)\right)^{-1}\right\}$$

*and* $\text{OPT} = \max_{\boldsymbol{a} \in \mathcal{A}} \sum_{t=1}^T r(\boldsymbol{a}_{:,t}, \boldsymbol{a}'_{:,t})$ *is the expected reward achieved by playing the best fixed interventions in hindsight. $\mathcal{B}$ and $\gamma_T$ are defined the same as in Theorem 1.*

*Proof.* We will find useful the notation of the regret of an individual agent $i \in [m]$. We will consider the regret of each agent to be not in terms of the reward $r$ but in terms of the feedback that agent receives: the UCB. We therefore define

$$R^i(T) = \max_a \sum_{t=1}^T \text{UCB}_t^{\mathcal{G}}(a, \boldsymbol{a}_{-i,t}, \boldsymbol{a}'_t) - \sum_{t=1}^T \text{UCB}_t^{\mathcal{G}}(a_{i,t}, \boldsymbol{a}_{-i,t}, \boldsymbol{a}'_t).$$

It can be thought of as the regret of agent $i$ compared to the best fixed action in hindsight, given that the actions of all other agents are fixed, and the agent is trying to maximize the sum of UCBs.

Using the above definitions, and provided that $\text{UCB}_t^{\mathcal{G}}(\cdot)$ are a valid upper confidence bound functions on the reward (according to Lemma 2), we can directly apply (Sessa et al., 2021, Theorem 1). This shows that with probability $1 - \frac{\delta}{2}$, the overall reward obtained by D-CBO-MW is lower bounded by

$$\sum_{t=1}^{T} r(\boldsymbol{a}_t, \boldsymbol{a}_t') \geq \alpha \cdot \text{OPT} - m \sum_{t=1}^{T} C_t \left( \frac{\delta}{2} \right) - \sum_{i=1}^{m} R^i(T),$$

where $\alpha$ is defined in Theorem 2 and $C_t(\delta/2)$ is as defined in Lemma 2. We note that Sessa et al. (2021) stated their theorem for the case where the UCB was computed using a single GP model (our setting with only a reward node and parent actions), however the proof is identical when any method to compute the UCB is used with an accompanying confidence bound in the form of Lemma 2.

Then, we can obtain the bound in the theorem statement by further bounding the agents' regrets $R^i(T)$. Indeed, because each agent updates its strategy (Line 10 in Algorithm 4) using the MW rule (Algorithm 3), we can bound each $R^i(T)$ with probability $1 - \frac{\delta}{2m}$ using Eq. (17). We can also substitute in for $C_t(\delta/2)$ using Lemma 2. Via a union bound we get our final result with probability $1 - \delta$. $\quad\square$

## C  EXPERIMENTS

### C.1  FUNCTION NETWORKS

We evaluate CBO-MW on 8 environments that have been modified from existing function networks Astudillo and Frazier (2021). These have been previously used to study CBO algorithms Sussex et al. (2022). We modify each environment in two ways (Perturb and Penny) in order to incorporate an adversary, resulting in 8 environments total. For all environments we tried to make the fewest modifications possible when incorporating the adversary in a way that made the game nontrivial while maintaining the spirit of the original function network.

Perturb environments allow the adversary to modify some of the input actions of the agents. Penny environments incorporate some element of the classic matching pennies game into the environment. If node $X_i$ has an adversary action parent, part of the mechanism for generating $X_i$ will involve multiplying another parent by the adversary action. Because the adversary can select negative actions (see below), this results in a dynamic similar to matching pennies.

Throughout the setup and theory we assume that there is one action per node for simplicity. For many of the function networks experiments there may be 0 or more than one action per node. Similar theoretical results for this case can also be shown using our analysis.

In all environments we map the discrete action space $[0, K - 1]$ to the continuous domain of the original function network. Using $a$ as the continuous action at a single node and $\bar{a}$ as the discrete action input at that node, we always use mapping $a = \left( \frac{\bar{a}}{K-1} - 0.5 \right) C_1 + C_2$, where $C_1, C_2$ defines some environment specific linear map that usually maps to the input space of the original function network in Astudillo and Frazier (2021). We use the same mapping for adversary actions in Perturb environments

For the adversary's actions in Penny environments, we use a more complicated mapping from $\bar{a}'$ to $a'$ to ensure that the adversary normally cannot select 0, which would result in a trivial game that the adversary can solve by always playing $\boldsymbol{a}' = \boldsymbol{0}$. If $K$ is even we use

$$a' = \left( \frac{\bar{a}'}{K-1}(1 - 2\epsilon)\epsilon - 0.5 \right) C_1 + C_2,$$

where $\epsilon = 0.05$. If $K$ is odd we use

$$a' = \left( \frac{\bar{a}' + 0.5}{K-1}(1 - 2\epsilon)\epsilon - 0.5 \right) C_1 + C_2,$$

where $\epsilon = 0.05$.

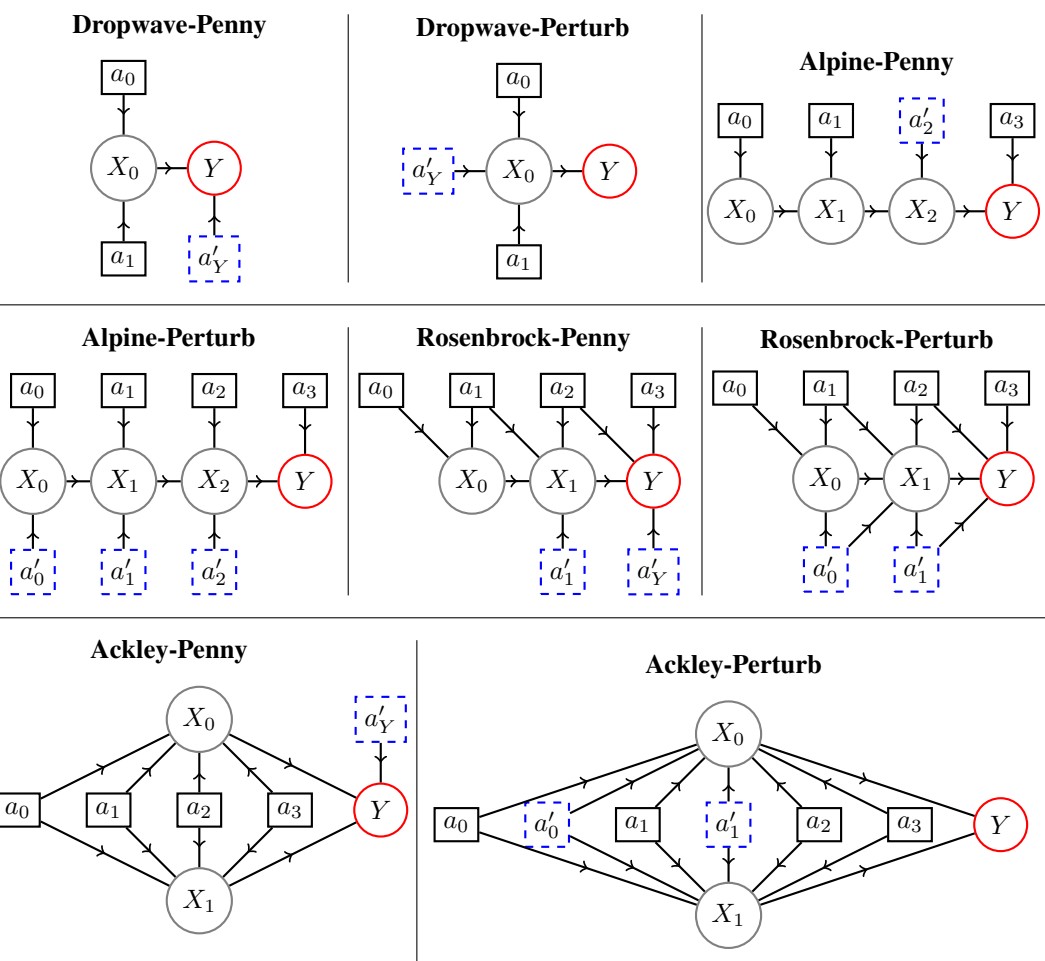

Figure 6: The DAGs corresponding to each task in the function networks experiments.

DAGs illustrating the casual structure of each environment are shown in Fig. 6. For some environments, we made the number of nodes smaller than the environment's counterpart in Astudillo and Frazier (2021) for computational reasons. We give the full SCM for each environment at the end of this subsection.

To select hyperparamaters for each method we perform a small hyperparameter sweep to select the best hyperparameters, before fixing the best hyperparameters and running 10 repeats on fresh seeds. On all repeats the agent is initialized with $2m + 1$ samples of uniformly randomly sampled $\boldsymbol{a}$ and $\boldsymbol{a}'$, where $m$ is the number of action nodes.

Fig. 7 shows the regret plots for environments not already shown in the main paper. As discussed, we find that standard non-adversarial BO algorithms often fail to achieve sublinear regret and that CBO-MW is often the most sample efficient compared to GP-MW. Only on Ackley-Penny is GP-MW obtaining a lower regret. This can be understood by our theory. Ackley-Penny is not at all sparse. That is, $\Delta \approx m$. Our Theorem 1 suggests that most improvement will come in high dimensional settings with sparse graphs. This is made clear by the superior performance of CBO-MW on the Alpine2 environments.

Here we systematically list the SCM for each environment.

**Dropwave-Penny**  $\boldsymbol{a} \in [0, 2]^2$, $\boldsymbol{a}' \in [-1, 1]^1$. We have

$$X_0 = \|\boldsymbol{a}\|,$$

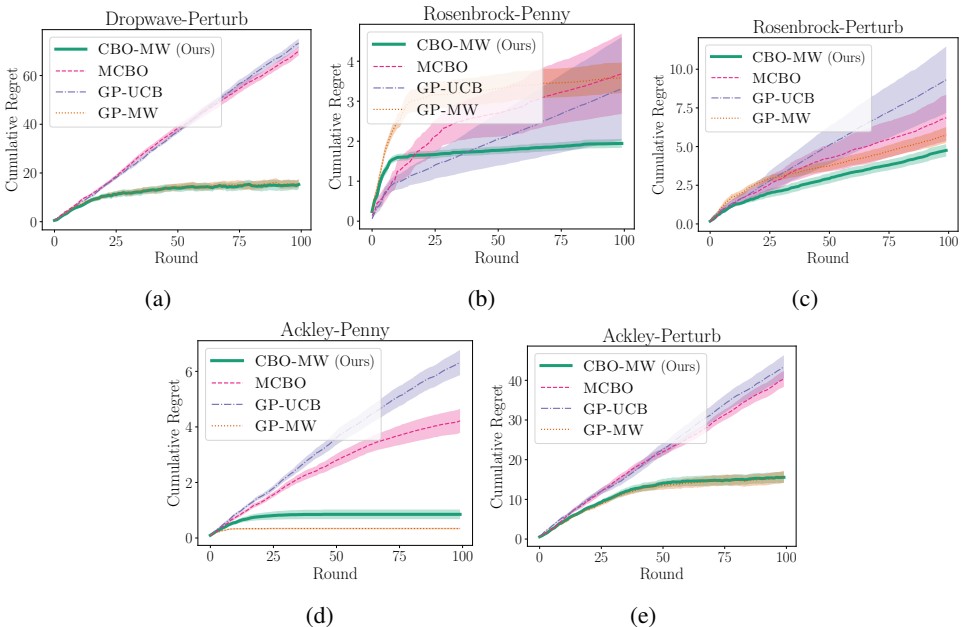

Figure 7: Regret plots for the function networks not already shown in Fig. 2.

$$Y = \frac{\cos(3X_0)}{2 + 0.5X_0^2} a_0'.$$

This is the original Dropwave from Astudillo and Frazier (2021) but with a modified input space and a matching pennies dynamic on the final node.

**Dropwave-Perturb**  $a \in [-10.24, 10.24]^2$, $a' \in [-10.24/5, 10.24/5]^1$. Like many Perturb systems, the adversary has a smaller domain than the agent to prevent it from being too strong. We have

$$X_0 = \left\| \left( \begin{bmatrix} a_0 - a_0' \\ a_1 \end{bmatrix} \right) \right\|,$$

$$Y = \frac{\cos(3X_0)}{2 + 0.5X_0^2}.$$

This is the original Dropwave from Astudillo and Frazier (2021) but with one of the actions being perturbed by the adversary.

**Alpine-Penny**  $a \in [0, 10]^4$, $a' \in [1, 11]^1$. We have
$$X_0 = -\sqrt{a_0} \sin(a_0)$$

For nodes $X_i$ with an adversary parent we have
$$X_i = -\sqrt{a_i'} \sin(a_i') X_{i-1},$$

and for nodes influenced by the agent we have
$$X_i = -\sqrt{a_i} \sin(a_i) X_{i-1}.$$

This is the original Alpine2 from Astudillo and Frazier (2021) but with an adversary controlling one of the nodes instead of the agent. We shift that adversary's action space so that they cannot 0 the output with a fixed action.

**Alpine-Perturb**  $a \in [0, 10]^4$, $a' \in [0, 2]^3$. Let $\bar{a}_i = a_i + a_i'$ for $i$ when $X_i$ has an adversarial action input, and $\bar{a}_i = a_i$ otherwise. We have
$$X_0 = -\sqrt{\bar{a}_0} \sin(\bar{a}_0),$$
$$X_i = -\sqrt{\bar{a}_i'} \sin(\bar{a}_i) X_{i-1}.$$

**Rosenbrock-Penny** $\boldsymbol{a} \in [0,1]^4$, $\boldsymbol{a}' \in [0,1]^2$. We have

$$X_0 = -100(a_1 - a_0^2)^2 - (1 - a_0)^2 + 10,$$

$$X_i = \left(-100(a_{i+1} - a_i^2)^2 - (1 - a_i)^2 + 10 + X_{i-1}\right) \bar{a}_i',$$

where $\bar{a}_i' = 1$ if there is no adversary over node $i$ and otherwise $\bar{a}_i' = a_i'$.

**Rosenbrock-Perturb** $\boldsymbol{a} \in [-2,2]^4$, $\boldsymbol{a}' \in [-1,1]^2$. Let $\bar{a}_i = a_i + a_i'$ for $i$ when $X_i$ has an adversarial action input, and $\bar{a}_i = a_i$ otherwise. We have

$$X_0 = -100(\bar{a}_1 - \bar{a}_0^2)^2 - (1 - \bar{a}_0)^2 + 10,$$

$$X_i = -100(\bar{a}_{i+1} - \bar{a}_i^2)^2 - (1 - \bar{a}_i)^2 + 10 + X_{i-1}.$$

**Ackley-Penny** $\boldsymbol{a} \in [-2,2]^4$, $\boldsymbol{a}' \in [-1,1]^1$. Let $\bar{a}_i = a_i + a_i'$ for $i$ when $X_i$ has an adversarial action input, and $\bar{a}_i = a_i$ otherwise. We have

$$X_0 = \frac{1}{4} \sum_i \bar{a}_i^2,$$

$$X_1 = \frac{1}{4} \sum_i \cos(2\pi \bar{a}_i),$$

$$Y = 20 \exp(-0.2\sqrt{X_0}) + e^{X_1}.$$

**Ackley-Perturb** $\boldsymbol{a} \in [-2,2]^4$, $\boldsymbol{a}' \in [-1,1]^2$. We have

$$X_0 = \frac{1}{4} \sum_i a_i^2,$$

$$X_1 = \frac{1}{4} \sum_i \cos(2\pi a_i),$$

$$Y = 20 a_0' \exp(-0.2\sqrt{X_0}) + e^{X_1}.$$

This is the original Ackley from Astudillo and Frazier (2021) but with a matching pennies dynamic on the final node.

## C.2 Shared Mobility System

We use the same SMS simulator as Sessa et al. (2021) and thus we largely refer to this regarding simulator details, unless otherwise specified. The simulator uses real demand data amalgamated from several SMS sources in Louisville Kentucky Lou (2021). We treat the system as a single SMS where all transport units are identical. A single trip taken in the Louisville data at a specific time is treated as a single unit of demand in the simulator. The demand is fulfilled if the location of the demand (where the trip started in the dataset) is within a certain distance (0.8km Euclidean distance) of a depot containing at least one bike. If the demand for a trip is satisfied, a single trip starting from the depot is counted and the bike is transported to the depot nearest to the end location of the trip in the dataset, after removing the bike from the system for 30 minutes to simulate the trip having non-zero duration. The simulator's use of real trip data means that the geographical and temporal distribution of demand, and its relation to the weather, is realistic.

The depots are not in the original trip data but constructed from the trip data using a similar procedure to Sessa et al. (2021). The start location for every trip taken over the year is clustered via k-means, and then clusters that are very close together are removed. This left 116 depots where bikes can be left. We consider a system with 40 bikes, which are distributed initially by 5 trucks that place all bikes in that truck at the same depot.

We further allocate depots to regions. These are constructed by using trip data across the whole year, and using a heuristic that clusters depots into regions so that there is a low chance that any given trip starts in one region and ends in another. As shown in Fig. 8 this leads to nearby depots often being in the same region, which is reasonable. We get 15 regions $R_1$ to $R_{15}$.

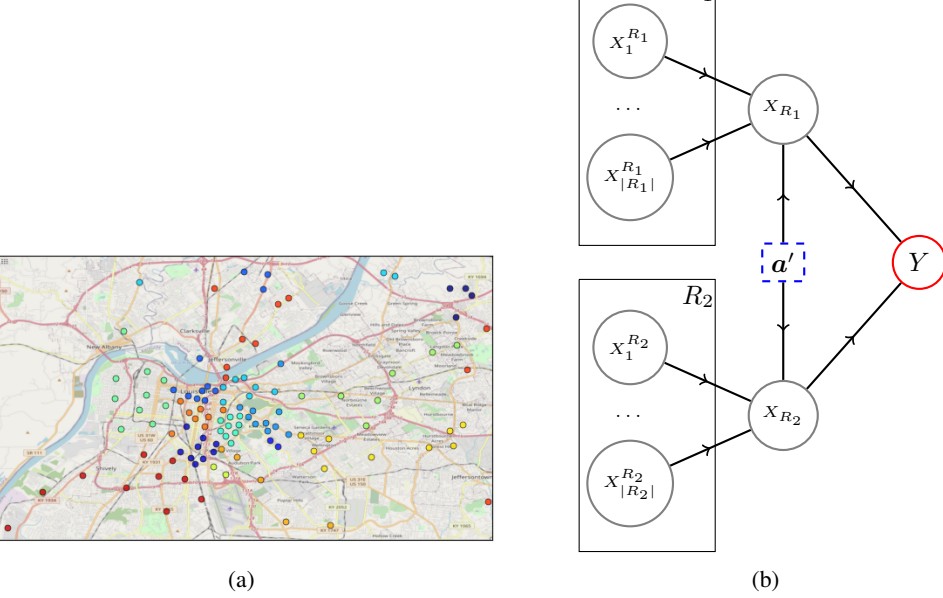

(a)             (b)

Figure 8: **(a)** The allocation of depots to regions. Depots of the same colour belong to the same region. **(b)** For our empirical evaluation of D-CBO-MW we use a graph that computes trips for each region and then sums these up to get total trips. Here we show a simplified version for 2 regions $R_1$ and $R_2$. The total trips in the first region $X_{R_1}$ only depends on the number of bikes allocated to each depot in $R_1$ given by $X_1^{R_1} \ldots X_{|R_1|}^{R_1}$. This reduces the dimensionality of GPs used in the model. For simplicity we have removed the agent's action nodes from the graph since the relationship between actions and bike allocations is a fixed known mapping.

Agent action $\boldsymbol{a}_i$ is a one-hot 116-length vector for which depot truck $i$'s bikes are placed at.

We obtain 3 measurements for $\boldsymbol{a}_t'$ at the end of each day $t$. This is the day's average temperature, rainfall, and total demand (including unmet demand). These are part of $\boldsymbol{a}'$ because they are out of our agent's control and not sampled *i.i.d.* across days. The agent must adaptively respond to these observations over time. Weather data is the real weather from that day obtained from Loc.

Observations $\boldsymbol{x}_i^r$ give the number of bikes at day start in depot number $i$ within region $r$. $\boldsymbol{x}_r$ is the total fulfilled trips that started in region $r$. Reward $Y$ is then the total trips in a day. All observations are normalized to ensure they are fixed in $[0, 1]$.

In Sessa et al. (2021), 2 separate GPs are used to model weekday and weekend demand. For simplicity we use a simulator that skips weekends, and therefore we don't need a separate model for the two types of day. No matter the algorithm used, the first 10 days of the year use the RANDOM strategy to gather exploratory data to initialize the GP models for the $\boldsymbol{x}_r$.

The graph used by D-CBO-MW is given in Fig. 8(b). The relationship between the bike allocations and bike distribution at day start $\{X_{:}^r\}_{r=R_1}^{R_{15}}$ is a fixed known function. The mechanism from the starting bike distribution in a region $r$ ($X_{:}^r$), adversary actions $\boldsymbol{a}'$ (weather and demand) and total trips in region $r$ over the day ($X_r$) is an unknown function that must be learnt for each $r$. The relationship between total trips $Y$ and its parents is known (sum over parents). For this kind of graph the output of the Causal UCB Oracle (Algorithm 2) will always set $\eta = 1$, because more trips in any region results in more total trips. For computational efficiency we therefore implement the Causal UCB Oracle to set $\eta$ to $\boldsymbol{1}$ instead of optimising over $\eta$.

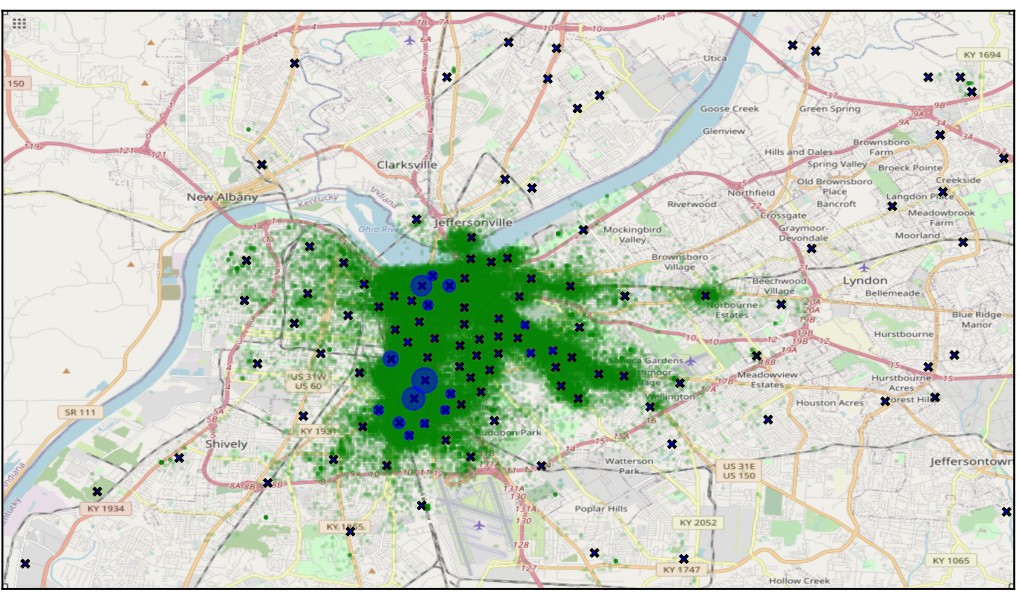

Figure 9: Green dots represent demand for trips, black crosses are depot locations, and blue circles are scaled in size to the frequency that D-CBO-MW assigns bikes to that depot (averaged across all repeats).

