# OpenReview forum: "Adversarial Causal Bayesian Optimization"
_ICLR.cc/2024/Conference — ICLR 2024 poster_

### Official Review · Reviewer_VBSi · 2023-10-25

**Soundness:** 3 good
**Presentation:** 3 good
**Contribution:** 3 good
**Rating:** 6
**Confidence:** 4

**Summary:**

The authors present a method for causal Bayesian optimization in non-stationary where the authors also allow for multi-agent environments. They present result on eight synthetic environment and one (very interesting) real environment where they demonstrate competitive results.

**Strengths:**

The main review, with comments and questions, are all in this section owing to the flow in which this review was conducted.

### Abstract

- I think there is some ambiguity w.r.t. the first sentence: the DAG can be known but the relationships (the mechanisms) of the DAG unknown, or vice versa. Which one do you mean? In the original paper (Aglietti et al) the DAG was always assumed known but there are other settings where the DAG is assumed unknown.
- Good abstract. Perhaps a bit more information on the experiments you performed (just the one sentence ought to do it).

### Introduction

- The supply example is great, but the way you introduce it in the second paragraph, reads a bit forced. Consider rephrasing. It doesn't sound very good at the moment.
- If you are modelling a phenomena in an environment that is changing, why is it not possible to model that with a temporal DAG? Or dynamical DAG? There is plenty of that type of work being done.
- Fig 1c - I am confused. Your blue nodes sound very much like standard non-manipulative variables and the idea of non-manipulative variables in causal setting, was introduced a long time ago. How are your blue nodes different?

### Background and problem statement

- There is some confusion here. In the abstract you said that $\mathcal{G}$ was unknown and now at the start of paragraph one you say that $\mathcal{G}$ is in fact know. Which is it?
- Is there a reason you deviate from the standard SCM definition from Pearl with $\langle U,V,P,F \rangle$? It seems unnecessary to introduce new notation for a setting which is well defined and well studied. You are just saying, in different words and notation, the interaction between the endogenous and exogenous variables in the SEM. More confusingly though you say that the $\mathcal{G}$ is part of your SCM definition whereas in the standard setting (well, Pearl's setting) the causal diagram is induced by the SCM, not part of it. See chapter 3 (Pearl, 2009).
- What is the reasoning behind using soft rather than hard interventions? What would happen if you used hard instead?
- There are as many actions as there are nodes $m$ in the graph? But then you are also intervening on the reward variable which is non-manipulative?
- To check my understanding: actions are continuous, but there are a finite amount of continuous actions, the cardinality of the domain of each action is then $K$? Why isn't each action just continuous?
- I think you should rephrase the uncertain parts of your problem statement: it is not the case that the causal model is unknown (this typically means the graph) but rather that the mechanisms of the SCM are unknown. You are not being precise enough at the moment to ward off ambiguity. Please change.

### Method

- I think this very important part deserves a deeper treatment, you say "Contrary to standard CBO (where algorithms can choose actions deterministically), in adversarial environments such as ACBO randomization is necessary to achieve no-regret" - why is that the case? Are you then saying that if you are using deterministic action selection it would be impossible to attain no-regret?
- Consider using left-pointing arrows in algorithm 2 to make it more procedural, in place of using equality signs on line 4 and 5. That goes in general for all your algorithms.
- How many times do you have to initialise the neural networks in algorithm 2 for this to work?

### Analysis

- It would be helpful if you gave an example of a Lipschitz continuous kernel, for uninitiated reader. I would also like to know what the consequence would be if you did not make this continuity assumption and how realistic it is?

### Computational considerations in larger action spaces

- Can you please comment on this line: "even with a large number of action variables $m$, $|\mathcal{A}|$ may still be small and thus CBO-MW feasible to implement" - what is 'large' here? When does it become unfeasible? Some numerical ballpark figures would be helpful.

### Experiments

- To confirm: you are considering the causally sufficient setting i.e. you assume there are no unobserved confounders? If so, please state that early on in the paper (apologies if I missed it).
- Would it also make sense to also compare against CBO? Don't worry this review is not conditional upon you doing that, I am merely wondering why it is not part of your analysis, seeing as you talk about it early on.
- The SMS example is _great_. Really enjoyed reading that.

**Weaknesses:**

See strengths.

Note: I have given this is a five to start with. I would be happy to increase my score following author engagement.

**Questions:**

See strengths.

---

> ### Author Response · Authors · 2023-11-17
> **Response to reviewer VBSi (part 1/2)**
>
> We thank the reviewer for their thorough review.
>
> **I think there is some ambiguity w.r.t. the first sentence: the DAG can be known but the relationships (the mechanisms) of the DAG unknown, or vice versa. Which one do you mean? In the original paper (Aglietti et al) the DAG was always assumed known but there are other settings where the DAG is assumed unknown.**
>
> We know the graph $G$ but not that mechanism (the $f_i$). The same as Aglietti et al. We think that this is clear from the paper itself. We can make this more explicit in the abstract by changing this sentence to “... on a structural causal model with known graph but unknown mechanisms…”.
>
> **If you are modelling a phenomena in an environment that is changing, why is it not possible to model that with a temporal DAG? Or dynamical DAG? There is plenty of that type of work being done.**
>
> Here we model a changing environment where we measure and react to potentially adversarial changes. Our understanding is that the use of a temporal or dynamical DAG requires an explicit probabilistic model for the changing environment, whereas we do not need to create a model for the adversary’s behavior. We believe that it is not obvious how one could concretely use temporal and dynamical DAG models in our setup and with the notion of regret we study.
>
> **Your blue nodes sound very much like standard non-manipulative variables**
>
> Assuming the reviewer refers to the non-manipulable variables studied in [1],  these are variables in the graph that one cannot intervene on. As we describe in section 2, we consider a soft intervention model where the $a_i’$ are parameters of the soft intervention on observation $X_i$. Therefore, we don’t see a connection between $a’$ and non-manipulative variables, because the former are the parameters of an intervention (performed by an adversary).
>
> [1] Lee, S., & Bareinboim, E. (2019). Structural Causal Bandits with Non-Manipulable Variables. Proceedings of the AAAI Conference on Artificial Intelligence, 33(01), 4164-4172.
>
> **What is the reasoning behind using soft rather than hard interventions? What would happen if you used hard instead?**
>
> Thanks for your question. We note that one could still run CBO-MW  with hard interventions without modifications. This would however be incompatible with some of the technical assumptions in the analysis (that all $\mu_i, \sigma_i$ are Lipschitz). Hence we have focused on the soft intervention model for simplicity.
>
> **There are as many actions as there are nodes m in the graph? But then you are also intervening on the reward variable which is non-manipulative?**
>
> We note that the learner is not necessarily allowed to intervene on the reward, in which case $\mid \cal{A_m} \mid = 1$ and all our results still go through. However, a soft intervention model generally allows the agent to directly intervene on the reward. Since soft interventions don’t break the causal relationship between a variable and it’s parents, it is still meaningful to consider cases where the reward can be intervened upon.
>
> **To check my understanding: actions are continuous, but there are a finite amount of continuous actions, the cardinality of the domain of each action is then K? Why isn't each action just continuous?**
>
> The reviewer is correct. The reason why we consider finitely many actions, though, is for the learner to tractably achieve no regret. Indeed, in the adversarial setting, the learner must randomize its decision and sample actions from a mixed strategy. Such a strategy is tractable to compute, and update  (line 6 of algorithm 1), if there is a finite number of actions. In practice, however, continuous action spaces can be discretized.
>
> **"Contrary to standard CBO (where algorithms can choose actions deterministically), in adversarial environments such as ACBO randomization is necessary to achieve no-regret" - why is that the case?**
>
> Intuitively, because the adversary has access to the game history and to the learner’s algorithm, if such an algorithm was deterministic then the adversary could anticipate the learner’s moves and thus inflict positive regret at each round. This is a standard argument in adversarial online learning and multiplayer games.
>
> **How many times do you have to initialise the neural networks in algorithm 2 for this to work?**
>
> In our experiments we performed a single initialization.
>
> **It would be helpful if you gave an example of a Lipschitz continuous kernel**
>
> Many commonly used kernels over continuous domains are Lipschitz continuous. Two examples: linear kernel and squared exponential kernel. This is mentioned in the cited work that we defer to in our paper.
>
> **what the consequence would be if you did not make this continuity assumption**
>
> We note that such an assumption is only required for the analysis of our regret guarantee. It is not a required assumption to apply the algorithm in practice.

---

> > ### Author Response · Authors · 2023-11-17
> > **Response to reviewer VBSi (part 2/2)**
> >
> > **Can you please comment on this line: "even with a large number of action variables m,  $\mid \cal{A} \mid$ may still be small and thus CBO-MW feasible to implement" - what is 'large' here?**
> >
> > The computational complexity of Algorithm 1 is O(K^m) - see line 6 of algorithm 1 -  where K is the number of possible values each action node can take. The motivation of section 6 is to have an algorithm with computational complexity linear in m.
> >
> > **To confirm: you are considering the causally sufficient setting i.e. you assume there are no unobserved confounders?**
> >
> > Thanks for pointing this out. Indeed we consider the case of no unobserved confounders. We will add a note early on to make this clear.
> >
> > **Would it also make sense to also compare against CBO?**
> >
> > Firstly, CBO assumes hard interventions so cannot be directly applied at least to the experimental settings we consider. We compare against MCBO on the synthetic experiments. In the MCBO paper, MCBO is compared to CBO with hard interventions in the stochastic BO setting under the cumulative regret metric, and MCBO performs relatively favorably.
> >
> > We hope the above points clarify the reviewer’s concerns and any misunderstandings. We are happy to expand more based on the reviewer’ feedback.

---

> ### Author Response · Authors · 2023-11-21
> **Reminder - end of discussion period soon**
>
> We would like to ask the reviewer whether our review responses and the revision clarify the reviewer’s concerns and change their score considerations. We are happy to provide further clarifications while the platform is still open for discussion, which is until the end of 22nd of November.

---

> > ### Comment · Reviewer_VBSi · 2023-11-22
> > **Acknowledging**
> >
> > Thanks to the authors for a thorough response. I confirm that I have read their review and have no further questions or comments.

---

### Official Review · Reviewer_aRF4 · 2023-10-30

**Soundness:** 3 good
**Presentation:** 2 fair
**Contribution:** 2 fair
**Rating:** 6
**Confidence:** 2

**Summary:**

The paper studies a model where an agent interacts with an unknown causal model that is partly controlled by an adversary. The problem is formulated as a Bayesian optimization problem. The paper proposes an algorithm based on multiplicative weights to solve the problem, which also uses the idea of the upper confidence bound algorith, that adopts an optimistic view in the face of uncertainty. Regret bounds were derived in the paper, and the proposed algorithm was evaluated empirically.

**Strengths:**

The idea of studying an online causal model looks interesting.

**Weaknesses:**

I don't see any fundamental difference between the studied model and a standard bandit or Bayesian optimization problem, where part of the model is stochastic and part of it is controlled by an adversary. Therefore, apart from having a causal model in the story, the novelty of the contribution seems limited.

**Questions:**

- It is mentioned in the problem statement that the adversary does not know the action to be performed by the agent. Could you explain what the choice of the adversary's action is based on? If the worst-case analysis is applied here, does it matter whether the adversary know the agent's action or not since the adversary will always act in the worst way anyhow? Or does the adversary choose the worst action based only on history actions? But then do they know the agent's algorithm or not? The assumption that they don't know the agent's action seems a bit odd.

- Could you explain the difference between your model and a model that combines stochanstic and adversary bandit?

---

> ### Author Response · Authors · 2023-11-17
> **Response to reviewer aRF4**
>
> We thank the reviewer for their review. Below, we respond to the reviewer’s concerns.
>
> **Could you explain what the choice of the adversary's action is based on?**
>
> From the text: “We assume the adversary does not have the power to know $a_{:, t}$ when selecting $a′_{:,t}$, but only has access to the history of interactions until round $t$.” Additionally, yes, we assume the adversary can know the agent’s algorithm (we’ll add a note to make this clear in the text). Since the agent can play a mixed strategy, the adversary can at best know the distribution the agent will select actions from at each round. This is not as strong of an adversary as knowing the agent’s exact actions, and makes the problem more intricate as the learner can achieve no regret if it’s mixed strategy is updated suitably (as we propose in the CBO-MW algorithm). We remark that this is a standard setup for the adversary in adversarial online learning.
>
> If the adversary at time $t$ could see the agent’s action $a_t$ before selecting $a_t’$, it would be a Stackelberg game. We could also model this in our framework by directly modeling the adversary’s action as an observation $X_i$, since it is a direct reaction to the action we chose. We describe this in appendix A.1.1.
>
> **Could you explain the difference between your model and a model that combines stochanstic and adversary bandit?**
>
> Here we request further information from the reviewer. It is not clear to us what the point of comparison is because it is not obvious what it means to just combine a stochastic and adversarial bandit. We think if the claim is that the contribution is limited, it would be easier for us to give a helpful response if the reviewer 1) refers to specific works in the literature and then 2) describes how the delta between our work and these is trivial. We think this would allow us to more concretely discuss the contribution.
>
> **I don't see any fundamental difference between the studied model and a standard bandit or Bayesian optimization problem, where part of the model is stochastic and part of it is controlled by an adversary.**
>
> We respectfully disagree with the reviewer and we hope that the points discussed below can further clarify the novelty of our approach. In particular, we believe there is significantly novelty compared to previous Bayesian optimization methods:
> - Because some part of the model is adversarial, the learner must randomize its actions to achieve no regret. Although standard adversarial online learning algorithms (such as multiplicative weights) exist for this, they require full-information feedback which is not available in Bayesian optimization.
> - The only Bayesian optimization approach that can provably achieve no regret in such a setting is GP-MW. However, this algorithm does not build a causal graph and can thus be highly sample inefficient, as we show.
> - In light of the above, CBO-MW is the first Bayesian optimization algorithm which can: 1) achieve no regret using 2) the learning of a causal reward model.  Similar to GP-MW, CBO-MW also computes optimistic estimates for the full information feedback. However, because such estimates are obtained through computing a counterfactual using a causal graph, we provide a practical subroutine to compute these (optimistic) counterfactuals in an efficient way (section 4.3).
> - We provide a regret guarantee and can show significantly improved rates compared to GP-MW  (section 5)
> - We show how to implement CBO-MW in a distributed way to improve the computational efficiency (section 6) and demonstrate good empirical performance (section 7).
>
> We hope the above points clarify the reviewer’s concerns. We are happy to expand more based on the reviewer’ feedback.

---

> > ### Comment · Reviewer_aRF4 · 2023-11-22
> > **Thank you for your responses**
> >
> > Thank you for your clarifications. I updated my score.

---

> ### Author Response · Authors · 2023-11-21
> **Reminder- end of discussion period soon**
>
> We would like to ask the reviewer whether our review responses and the revision clarify the reviewer’s concerns and change their score considerations. We are happy to provide further clarifications while the platform is still open for discussion, which is until the end of 22nd of November.

---

### Official Review · Reviewer_h5FK · 2023-10-31

**Soundness:** 3 good
**Presentation:** 3 good
**Contribution:** 2 fair
**Rating:** 6
**Confidence:** 4

**Summary:**

The paper studies causal bayesian optimization under certain kinds of adversaries who can pick additive variables in the causal graph post seeing the variables of the agent up to time t-1.

- They derive regret upperbounds for a variant of the multiplicative weights algorithm and show scaling with sqrt(T).
- The analysis is reasonably strong, but motivation could be made more clear -- the motivating examples are not necessarily adversarial.

**Strengths:**

- Experiments show that the algorithm is strong for the use cases considered.
- Well written problem statement.
- For the model chosen, the analysis is sound.

**Weaknesses:**

- The related work ignores causal bandit literature?
- The problem is not well motivated. Why is SMS adversarial and not stochastic?
- The graph notations are confusing. The typical graph has 1 root node. But in causal graphs, we may have multiple nodes without parents.
- The adversary cannot see the action taken by agent before taking its action. Is this adversary weak? You have considered that the agent can see adversary's action before choosing their own action set, but what about vice-versa?
- The additive term Beta^{N+1} does seem high.
- Note that "Causal Bandits for Linear Structural Equation Models" Varici et al 2023 show that the regret scales as length of the longest causal path in the graph for linear SCMs, whereas you consider N - length of path to root node. The former (not cited in your work), seems tighter.
- Is the assumption of usage of only finite action spaces chosen from continuous Reals_[0,1] feasible? If we draw an epsilon net over [0,1], then the computation complexity of Alg 1 may rise.
- There are exponential combination of adversarial choices, for each of which a counterfactual computation may be taken up. This is computationally demanding.

**Questions:**

- Page 2: If X_m is a leaf, and it is the reward variable, then it has no parents? Did you mean X_0 is the reward variable?
- You speak of Adversarial CBO, but assume a SCM. Do Causal Bayesian Networks involve the functional relations between the variables?
- Why does cumulative regret go down with increasing rounds for Dropwave Penny in Figure 2?
- What is the lower bound for regret?
- You speak to a sqrt(t) dependence on regret, but the regret curve flattens for your experiment (and even decreases) in the graphs. Why do you believe this is happening?
- Notation question: fi: Zi × Ai × A′i → Xi. Should this not be fi: Zi × Ai × A′i × Ω → Xi
- You say "Because mechanisms can be non-monotonic and nonlinear, one cannot simply independently maximize the output of every mechanism. We thus defer this task to an algorithmic subroutine (denoted causal UCB oracle)". In this algorithm, you use a neural network explicitly. Does the error in functional approximation due to nn use not flow into the regret term?
- Can line 7 in Alg 1 be amended to optimize over a' in [0,1] as well?
- Why is it necessary to learn the causal function at each node, and not just at node Y, or at parents of Y? To bound reward estimates at Y, do we need equally good estimates at all nodes in the graph? (If not the search space for a,a' goes lower and therefore the number of calls to Alg2).


## Typos:
- Page 2 - "for the parents this node"

## Suggestions
- Please expand on the literature review.

---
Note: May be willing to improve the score based on author responses.

---

> ### Author Response · Authors · 2023-11-17
> **Response to reviewer h5FK (part 1/3)**
>
> We thank the reviewer for their thorough review.
>
> **“The related work ignores causal bandit literature?”**
> After the related work sentence “Agliettiet al.(2020) propose the first CBO setting with hard interventions and an algorithm that uses the do-calculus to generalise from observational to interventional data”, we will add “The CBO line of work builds off the causal bandit setting [cite Lattimore], which similarly incorporates causal graph knowledge into the bandit setting usually considering discrete actions with categorical observations or linear mechanisms with continuous observations [cite Varici]..”.
>
> **The problem is not well motivated. Why is SMS adversarial and not stochastic?**
>
> We mention this in the experiments: “After each day,we observe weather and demand data from the previous day which are highly non-stationary” but we will add a sentence to the introduction that makes this clear from the beginning. In the 4th paragraph we will add: “In the SMS application, we observe the demand and weather which can only be observed at the day’s end, and are highly non-stationary. For example, the weather distribution will vary seasonally.”
>
> **“The graph notations are confusing. The typical graph has 1 root node. But in causal graphs, we may have multiple nodes without parents.”**
> We were not able to follow this comment. If you could expand on it we are happy to take a look. Based on one of your later comments (addressed below), we think you might be getting root and leaf node mixed up.
>
> **“The adversary cannot see the action taken by agent before taking its action. Is this adversary weak? You have considered that the agent can see adversary's action before choosing their own action set, but what about vice-versa?”**
> If the adversary at time $t$ could see our action $a_t$ and then respond with $a_t’$, it would be a Stackelberg game and not a simultaneous action game. Our setting, algorithm, and guarantees can actually also model this Stackelberg game setting. Since the adversary would respond directly to our action $a_t$, we could model the adversary action as a function of $a_t$ (and therefore treat it as just another $X$ in the graph). We discuss this in appendix A.1.1.
>
> **“You have considered that the agent can see adversary's action before choosing their own action set”**
> This is not true. We consider a setting where the agent selects $a_t$ simultaneously with the adversary selecting $a_t’$, so the agent does not observe the adversary action beforehand. We believe that this is unambiguous in the paper from section 2 under “Problem Statement”.
>
> **“The additive term Beta^{N+1} does seem high.”**
> After the regret guarantee we compare our regret guarantee with a non-causal approach and give a strong case for why we have a significant improvement. We are also to our knowledge the first to give any kind of guarantee at all for this particular setting (besides the guarantee you get from applying GP-MW which uses no causal information).
>
> **“Note that "Causal Bandits for Linear Structural Equation Models" Varici et al 2023 show that the regret scales as length of the longest causal path in the graph for linear SCMs, whereas you consider N - length of path to root node. The former (not cited in your work), seems tighter.”**
>
> We think that it is not meaningful to directly compare the guarantees from our paper and Varici et al.. Here are the two key differences in the setups:
>
> - They consider linear mechanisms. We consider much more general models for each mechanism where they can be modeled with a GP (come from an RKHS with bounded norm). Linear models are closed under composition, so for their model the reward is still a linear function of the actions. For our setting, GPs are not closed under composition, so we get this emergent complexity where the reward as a function of actions is a much more complicated function class than just a single GP (it is a deep GP [1]).
> - They consider the stochastic setting and therefore a much “easier” notion of regret than the one we consider in the adversarial/non-stationary case (see equation 3).
>
> [1] Damianou, Andreas, and Neil D. Lawrence. "Deep gaussian processes." Artificial intelligence and statistics. PMLR, 2013.
>
> We compare our regret bound to the only baseline we know of that can achieve a guarantee in this setting (GP-MW) and give an argument for why our guarantee is favorable. Varici et al 2023 considers a less general problem and while we think it is a great result, it is for a different setting to our result.
>
> We’ll add a citation of Varici et al 2023 to the related work (see comment above on causal bandits).

---

> > ### Author Response · Authors · 2023-11-17
> > **Response to reviewer h5FK (part 2/3)**
> >
> > **“Is the assumption of usage of only finite action spaces chosen from continuous Reals_[0,1] feasible? If we draw an epsilon net over [0,1], then the computation complexity of Alg 1 may rise.”**
> >
> > Let's say that for each $\cal{A_i}$ we have the same finite number of possible actions $K$. Now you are correct that the complexity of Algorithm 1 will become large because the computational complexity is order of $K^m$, since we need to update a weight for every possible action combination. This is the point of D-GP-MW which we introduce in the paper. This algorithm also achieves a regret guarantee (section 6 plus associated appendix), and has computational complexity only linear in $K$ (if we have a separate agent controlling each $a_i$).  As a specific example we apply the algorithm to the SMS setting in the experiments and it works well.
> >
> > **“There are exponential combination of adversarial choices, for each of which a counterfactual computation may be taken up. This is computationally demanding.”**
> >
> > This is not true. In algorithm 1, at time $t$ we compute a counterfactual *fixing* $a_t’$ and therefore only iterate over $\cal{A}$ (not $\cal{A’}$). See section 4.2: “...is the counterfactual of what would have happened, in expectation over noise,had $a_t′$  remained fixed but the algorithm selected $a$ instead of $a_t$”. Therefore increasing the size of the adversary’s action space has no impact on the computational complexity of the algorithm.
> >
> > **“Page 2: If X_m is a leaf, and it is the reward variable, then it has no parents? Did you mean X_0 is the reward variable?”**
> >
> > A leaf is a node that has no children. $X_m$ is the reward variable and has no children.
> >
> > **“You speak of Adversarial CBO, but assume a SCM. Do Causal Bayesian Networks involve the functional relations between the variables?”**
> >
> > We assume that the data generating process can be written as an SCM as described in section 2, and we assume that our agent knows the causal graph but not the mechanisms between variables.
> >
> > **“Why does cumulative regret go down with increasing rounds for Dropwave Penny in Figure 2?”**
> >
> > This is possible because of the stronger notion of regret used in our work (see equation 3), compared to the notion used in Bayesian optimization with stochastic noise. Since the regret compares to the best that a single fixed action could have done (fixed over every time point), it is possible that at some time points (such as the most recent ones) the action our agent selects could outperform the best fixed action (because the best fixed action has to also be the best on average over the entirety of the actions the adversary selected so far).
> >
> > **“What is the lower bound for regret?”**
> > We don’t give a result for this and to our knowledge there are no results on lower bounds for causal BO regret in the literature (there are some for causal bandit), let alone for the adversarial case introduced here.
> >
> > **“You speak to a sqrt(t) dependence on regret, but the regret curve flattens for your experiment (and even decreases) in the graphs. Why do you believe this is happening?”**
> >
> > The guarantee is a worst-case upper bound. So it is not necessarily true that for every specific experiment this worst-case will be realized. We do find our theory is useful for predicting how regret will vary for different problems however.  From the paper: ”The setting where CBO-MW was not strongest involves a dense graph where worst-case GP- sparsity is the same as GP-MW, which is consistent with our theory. In settings such as Alpine where the graph is highly sparse, we observe the greatest improvements from using CBO-MW.” This is specifically happening because at some point in the experiments the agent begins behaving near-optimally
> >
> > **“Notation question: fi: Zi × Ai × A′i → Xi. Should this not be fi: Zi × Ai × A′i × Ω → Xi”**
> >
> > The original is correct since the noise is additive noise and not an input to $f_i$: see Equation 1.
> >
> > **“You say "Because mechanisms can be non-monotonic and nonlinear, one cannot simply independently maximize the output of every mechanism. We thus defer this task to an algorithmic subroutine (denoted causal UCB oracle)". In this algorithm, you use a neural network explicitly. Does the error in functional approximation due to nn use not flow into the regret term?”**
> >
> > We will make this clearer by adding a sentence into the analysis saying “For the analysis we assume access to a causal UCB Oracle is given to us. That is, we can always compute $UCB^G_t(a, a’)$ (equation 9) exactly.”  So our analysis does not account for possible error in computing the UCBs. When working in complex settings like ours (involving solving a non-convex optimization problem as a subroutine) it is common to make these types of assumptions. A relevant example is the use of an oracle in the MCBO analysis, an algorithm that we compare to.

---

> > > ### Author Response · Authors · 2023-11-17
> > > **Response to reviewer h5FK (part 3/3)**
> > >
> > > **Can line 7 in Alg 1 be amended to optimize over a' in [0,1] as well?**
> > >
> > > It is not clear to us why this would be necessary or what this modified algorithm would be achieving. We are happy to discuss if further details on this are given. If in line 7 we also optimize over $a’$, the weight updates never make any use of what the adversary actually did (it would only be used in model fitting), so we would not expect this algorithm to achieve sublinear regret since it is not able to adapt it’s strategy based on the actual actions the adversary is playing in the past.
> > >
> > > **Why is it necessary to learn the causal function at each node, and not just at node Y, or at parents of Y? To bound reward estimates at Y, do we need equally good estimates at all nodes in the graph? (If not the search space for a,a' goes lower and therefore the number of calls to Alg2).**
> > >
> > > If you just model Y given the actions and adversary actions, this is what GP-MW is doing and we compare with this in the analysis of the regret bound. The key idea is that if the graph has some level of sparsity you end up fitting several lower-dimensional models (one for each mechanism) instead of one higher-dimensional model (reward given all actions), and this trade-off ends up being favorable. You also end up with a richer function class since GPs are not closed under composition, as we discussed above. We don’t specifically analyze any algorithm that just model’s the parents of the reward but it seems possible that similar arguments to what we make for the "just at node Y" case could be made.
> > >
> > > We hope the above points clarify the reviewer’s concerns and any misunderstandings. We are happy to expand more based on the reviewer’ feedback.

---

> ### Author Response · Authors · 2023-11-21
> **Reminder - end of discussion period soon**
>
> We would like to ask the reviewer whether our review responses and the revision clarify the reviewer’s concerns and change their score considerations. For example, our clarification of relation to the causal bandit literature and relation of the regret bound to prior work. We are happy to provide further clarifications while the platform is still open for discussion, which is until the end of 22nd of November.

---

> > ### Comment · Reviewer_h5FK · 2023-11-22
> > **increasing score to 6**
> >
> > I have read your comments and thank you for the details. I increase my score to a 6.

---

### Official Review · Reviewer_xNam · 2023-10-31

**Soundness:** 2 fair
**Presentation:** 2 fair
**Contribution:** 2 fair
**Rating:** 6
**Confidence:** 4

**Summary:**

The authors propose an extension of the framework of CBO, named ACBO, where other agents (or external events) can also intervene on the system. This is to be able to model changes in the environment. The ACBO framework proposes a concrete algorithm to solve this problem, CBO-MW, which computes optimistic counterfactual reward estimates and enjoys cumulative regret bounds.

**Strengths:**

- Interesting setting of doing BO with causal relationships among variables and external interventions
- Interesting applied problem in the experiments

**Weaknesses:**

- The main weakness is the naming of the method, which refers to CBO (Aglietti et al 2020) and hereby its relationship with the CBO setting. The paper claims to show a generalization of CBO, which it seems to be an * algorithm * proposed in Aglietti et al 2020 as a solution to the "Causal Global Optimization" (CGO) problem. Here, in the abstract but also in the main paper, (1) no mention to the CGO problem is made (2) CBO seems to refer to the "setting" (somehow as a replacement to CGO), while the algorithm proposed here is CBO-MW. But CBO is not a setting, as said, it's an algorithm to solve the CGO under certain assumptions. In CBO, there are intervention *sets* and intervention *values/levels* (continuous-valued). I cannot see any of these here throughout, so it's unclear whether interventions on multiple variables here are excluded or what. In general, are you trying to also solve the CGO problem (a suitably modified version under external interventions, of course) or not ? Again on this point, the authors claim to compare using "previous CBO benchmarks", but there are no benchmarks actually from the CBO paper of Aglietti et al 2020, and the CBO-CW is actually *not* compared (nor experimentally, nor methodologically with a discussion) to CBO itself. There is also no "causal prior" associated to the GP as in CBO.
- My understanding then from the above is that this work is actually not an extension of CBO at all (although I would like to hear from the authors), rather an extension of GP-MW which is mentioned a lot and compared to experimentally.
- Strengthening my belief wrt to the "distance" of this work with CBO, the authors here evaluate performance with the **cumulative** regret. This does not seem to be used at all in CBO, where instead a simple regret seems to be used (Aglietti et al 2020).

**Questions:**

If the authors clarify significantly the relationship with CBO, and modify the claims and narrative in the paper accordingly, I am open to increasing my score.

---

> ### Author Response · Authors · 2023-11-17
> **Response to reviewer xNam**
>
> We thank the reviewer for their review.
>
> **Relation to CBO**
>
> We use ‘causal Bayesian optimization’ (CBO) to describe the general causal Bayesian optimization problem, which is separate from the specific algorithm proposed by Aglietti et al.  There have been several works studying this CBO setting since the original work of Aglietti et al. These works consider multiple variations of their setting: including soft intervention models (instead of hard interventions) and the cumulative regret metric (instead of the simulated regret metric). These are technical differences within a conceptually very similar set of problems.
>
> **The paper claims to show a generalization of CBO, which it seems to be an algorithm**
>
> Thank you for pointing out that in Aglietti et al. the authors specifically distinguish between causal global optimization (CGO) as their setting, and causal Bayesian optimization (CBO) as their algorithm. Therefore when specifically describing the setting of Aglietti et al 2019 we will refer to it as CGO in our paper too. We will however continue to refer to the general problem of global optimization in the presence of a causal graph and prior over mechanisms as causal Bayesian optimization. We will also refer to the setting of our paper as ACBO. This is consistent with the wider BO literature where BO - global optimization with a prior - is used to describe a setting and usually the algorithm is described based on the specific acquisition function e.g. UCB, UEI etc. We note that the paper of MCBO, a baseline we compare to, also describes their setting as ‘CBO’. The main difference here is whether you consider the prior as part of the setting or the algorithm. In Aglietti et al. they use a prior as part of their solution to the CGO problem, but in many other parts of the BO literature, the prior is considered part of the problem setting.
>
> **...CBO, there are intervention sets and intervention values/levels**
>
> CBO as in Aglietti et al. considers a hard intervention model. As discussed in the setup we consider a soft intervention model as in MCBO. Our model includes interventions on multiple variables. This explains the difference to what you read in Aglietti et al. Again we see this as more of a technical difference.
>
> **the authors claim to compare using "previous CBO benchmarks", but there are no benchmarks actually from the CBO paper of Aglietti et al 2020**
>
> Firstly, CBO assumes hard interventions so cannot be directly applied at least to the experimental settings we consider. We compare against MCBO on the synthetic experiments. In the MCBO paper, MCBO is compared to CBO with hard interventions in the stochastic BO setting under the cumulative regret metric, and MCBO performs favorably. We say “adversarial versions of previously studied CBO benchmarks”, meaning that we modify some of the environments from the MCBO paper (a paper about CBO) for the adversarial setting.
>
> We hope the above points clarify the reviewer’s concerns and any misunderstandings. We are happy to expand more on this based on the reviewer’ feedback.

---

> ### Author Response · Authors · 2023-11-21
> **Reminder- end of discussion period soon**
>
> We would like to ask the reviewer whether our review responses and the revision clarify the reviewer’s concerns and change their score considerations. In particular, our clarification over the naming of the algorithm and naming of the problem setting. We are happy to provide further clarifications while the platform is still open for discussion, which is until the end of 22nd of November.

---

> ### Comment · Reviewer_xNam · 2023-11-22
> **Response to authors**
>
> Thanks for engaging.
>
> Regarding the first two points, please make sure that readers will not be confused with these things. Yes the MCBO paper called CBO the problem as you said but it indeed confuses the literature. It would be better if you improved the literature and clarified things properly.
>
> **"Again we see this as more of a technical difference"**
>
> This technical difference makes a lot of difference. What do you mean by  "Our model includes interventions on multiple variables" ? Do you have to select which variables to intervene as well or it is given ? Again this is a major difficulty in CBO since of course there are a lot of subsets of the DAG variables to search over.
>
>
> "Firstly, CBO assumes hard interventions so cannot be directly applied at least to the experimental settings we consider"
>
> Again this is why it is confusing to say you are generalizing CBO throughout. I understand basically you want to use this name because it's simple and easy to remember, and I think it's even fine; but since there was something in the literature well defined and called CBO, it should be clear to the reader that "CBO" here is essentially not "CBO" of Aglietti et al 2020.
>
> The differences are not just technical but major (cumulative regret vs simple regret, no causal priors here like in CBO) and make a lot of difference in what the algorithms should be doing, what are the challenges etc., so they are basically "related" research areas but on a superficial level.
>
> With the hope that authors will clarify significantly the next version I will increase my score to 6.

---

> > ### Author Response · Authors · 2023-11-22
> > **Followup response to reviewer**
> >
> > Thank you for your response and constructive feedback.
> >
> > **"it is confusing to say you are generalizing CBO throughout."**
> >
> > We only explicitly use this wording in the abstract and it is true in the context of those sentences, since we do consider a more general version of "an agent intervenes on a structural causal model with known graph but unknown mechanisms to maximize a downstream reward variable". In the main body of the paper we believe that we give the more nuanced view that you are after:
> > "Aglietti et al.(2020) propose the first CBO setting with hard interventions and an algorithm that uses the do-calculus to generalise from observational to interventional data".  One thing that we would add here to the camera-ready would be to also note that Aglietti et al. considers unobserved confounding (which is related to why they need this causal prior that they use).
> >
> > **"cumulative regret vs simple regret"**
> > A sublinear cumulative regret implies a converging regret at any particular round. We believe that this is well understood in the BO literature.
> >
> > **Do you have to select which variables to intervene as well or it is given ?**
> > Yes this challenge can also be seen as a component of our setting. One can just consider one of the possible actions on each variable as the "not intervening on this variable" action.
> >
> > **"no causal priors here like in CBO"**
> > As mentioned above, we will add something to this sentence we quoted above to make it clear that Agliettti et al. are looking at a setting where unobserved confounding is present, which we hope would address your concern on this particular matter.
> >
> > We think that this minor change in the related work and our clarification here should be sufficient to address the raised concern. Please let us know if you agree and if it affects your score at all.

---

### Author Response · Authors · 2023-11-19
**Reviewer Feedback INcorporated into Submission**

Dear reviewers,

Thank you for taking the time so far to offer us constructive feedback on our paper. In some responses we noted that we would incorporate the suggestion into the paper to improve clarity. We have uploaded a new pdf with these additions in red. Note that the exact wording of the writing addition that we made may not be identical to the wording we used in the responses to reviews. We hope that these few additions help clarify any concerns.

The discussion window is only open for a few more days. We encourage reviewers to follow-up on our responses and let us know if their concerns have been addressed so that we are able to provide any additional input necessary before the discussion window ends.

Thank you!

---

### Meta-Review · Area_Chair_H3LM · 2023-12-07

**Metareview:**

The authors study a twist on the problem of causal Bayesian optimization wherein other (potentially adversarial) agents and events can intervene in the system of interest. The authors formalize this setting (calling it "adversarial causal Bayesian optimization"), provide an algorithm designed for this setting, prove a regret bound, and evaluate its performance in a series of experiments.

This appears to be a novel and interesting setting, and the authors provide a relatively complete package to start the conversation (formalization, algorithm, bound, experiments). I believe this work to be of interest to the ICLR community and that its inclusion to the program would be welcome.

The reviewers did note some minor issues in their initial reviews and the following discussion, mostly regarding clarity, notation, and motivation. I believe they provided some valuable suggestions throughout this process to help address these issues, and I strongly encourage the authors to take this discussion into account in revising the manuscript.

**Justification For Why Not Higher Score:**

Although the reviewers were unified in their recommendation of acceptance, the enthusiasm for the work was relatively muted and the support was nonetheless weak.

**Justification For Why Not Lower Score:**

Although support from the reviewers was not as strong as it possibly could have been, the authors are united in their judgement of the value of the manuscript. Further, I (along with the reviewers) judge the novelty of the problem formulation to be of independent interest and believe its discussion at ICLR would benefit the community.

---

### Decision · Program_Chairs · 2024-01-16

Accept (poster)